# Phenome-wide association studies across large population cohorts support drug target validation

Dorothée Diogo[1], Chao Tian[2], Christopher S. Franklin[3], Mervi Alanne-Kinnunen[4], Michael March[5], Chris C.A. Spencer[3], Ciara Vangjeli[3], Michael E. Weale[3], Hannele Mattsson[4,6], Elina Kilpeläinen[4], Patrick M.A. Sleiman[5], Dermot F. Reilly[1], Joshua McElwee[1,15], Joseph C. Maranville[1,16], Arnaub K. Chatterjee[1,17], Aman Bhandari[1,18], the 23andMe Research Team, Khanh-Dung H. Nguyen[7], Karol Estrada[7], Mary-Pat Reeve[8], Janna Hutz[8], Nan Bing[9], Sally John[7], Daniel G. MacArthur[10,11], Veikko Salomaa[6], Samuli Ripatti[4,10,12], Hakon Hakonarson[5], Mark J. Daly[10,11], Aarno Palotie[4,10,11,13,14], David A. Hinds[2], Peter Donnelly[3], Caroline S. Fox[1], Aaron G. Day-Williams[1,7], Robert M. Plenge[1,16] & Heiko Runz[1,19]

Phenome-wide association studies (PheWAS) have been proposed as a possible aid in drug development through elucidating mechanisms of action, identifying alternative indications, or predicting adverse drug events (ADEs). Here, we select 25 single nucleotide polymorphisms (SNPs) linked through genome-wide association studies (GWAS) to 19 candidate drug targets for common disease indications. We interrogate these SNPs by PheWAS in four large cohorts with extensive health information (23andMe, UK Biobank, FINRISK, CHOP) for association with 1683 binary endpoints in up to 697,815 individuals and conduct meta-analyses for 145 mapped disease endpoints. Our analyses replicate 75% of known GWAS associations ($P < 0.05$) and identify nine study-wide significant novel associations (of 71 with FDR < 0.1). We describe associations that may predict ADEs, e.g., acne, high cholesterol, gout, and gallstones with rs738409 (p.I148M) in *PNPLA3* and asthma with rs1990760 (p.T946A) in *IFIH1*. Our results demonstrate PheWAS as a powerful addition to the toolkit for drug discovery.

[1] Merck Sharp & Dohme, Boston, MA 02115, USA. [2] 23andMe Inc, Mountain View, CA 94041, USA. [3] Genomics plc, Oxford OX1 1JD, UK. [4] Institute for Molecular Medicine Finland (FIMM), University of Helsinki, 00014 Helsinki, Finland. [5] The Children's Hospital of Philadelphia and University of Pennsylvania, Philadelphia, PA 19104, USA. [6] National Institute for Health and Welfare, FI-00271 Helsinki, Finland. [7] Biogen, Research and Early Development, Cambridge, MA 02142, USA. [8] Eisai, Andover, MA 01810, USA. [9] Pfizer, Cambridge, MA 02139, USA. [10] Broad Institute of MIT and Harvard, Cambridge, MA 02142, USA. [11] Analytic and Translational Genetics Unit, Department of Medicine, Massachusetts General Hospital, Boston, MA 02114, USA. [12] Department of Public Health, University of Helsinki, Helsinki, Finland. [13] Psychiatric & Neurodevelopmental Genetics Unit, Department of Psychiatry, Massachusetts General Hospital, Boston, MA, USA. [14] Department of Neurology, Massachusetts General Hospital, Boston, MA 02114, USA. [15] Present address: Nimbus Therapeutics, Cambridge, MA 02139, USA. [16] Present address: Celgene, Cambridge, MA 02140, USA. [17] Present address: McKinsey & Co., Boston, MA 02210, USA. [18] Present address: Vertex Pharmaceuticals, Boston, MA 02210, USA. [19] Present address: Biogen, Research and Early Development, Cambridge, MA 02142, USA. A full list of consortium members appears at the end of the paper. Correspondence and requests for materials should be addressed to D.D. (email: dorothee.diogo@merck.com) or to H.R. (email: heiko.runz@gmail.com)

The discovery and development of novel therapeutics is difficult. It may take 15 years to advance a new molecular entity from therapeutic hypothesis to approval, with development costs in the billion dollar range and only a 10% chance of a new drug tested in humans eventually getting approval[1]. Two reasons stand out to explain the high failure rate of clinical trials and receding return on R&D investment across the pharmaceutical industry: a lower efficacy of the compound in the targeted disease population than anticipated from preclinical studies; and the occurrence of unintended drug effects, particularly mechanism-based adverse drug events (ADEs) uncovered only in late-stage clinical trials[2]. A greater understanding of human data relevant to the drug target at early stages of drug development is generally considered to increase the probability of success[1,3,4].

Resources that systematically capture biomedical information on vast numbers of individuals are revolutionizing our ability to understand the complexities of human biology and morbidity. Electronic health records (EHRs) and other resources that systematically capture extensive health information have rapidly become well-established tools for epidemiological and post-marketing research[5,6]. Recently, a surge of initiatives are seeking to link such phenotype resources with genome-scale genetic data in order to gain further insights into the genetics of common diseases[7–14].

An attractive approach to help accelerate drug development utilizing these genotype–phenotype resources is through applying phenome-wide association studies (PheWAS). PheWAS are an unbiased approach to test for associations between a specific genetic variant, or, more recently, combination of variants, and a wide range of phenotypes in large numbers of individuals[7,15,16]. By exploring the associations of a genetic variant that impacts the function of a drug target gene, PheWAS in disease-agnostic cohorts with extensive health information may enrich the drug discovery process for five reasons: (1) association studies in disease-agnostic cohorts may validate target-disease links in cohorts that more closely resemble the real-world, i.e., the patients that will ultimately receive a drug;[17] (2) by unraveling pleiotropy, PheWAS may improve our understanding of the biological functions of a target, or hint at concealed pathophysiological connections between disease entities previously considered as distinct;[18,19] (3) PheWAS may reveal opportunities for drug repurposing, an attractive alternative to de novo drug development;[20,21] (4) PheWAS may point to phenotypes that associate with an inverse directionality of target function, thus unraveling potential ADEs at very early stages of a developmental program, minimize risks to trial participants, and help define the most appropriate patient populations to benefit from a drug;[21] and (5) through quantitative estimates from genetic safety and efficacy profiles, PheWAS may help prioritize multiple possible targets by identifying the target with the most promising therapeutic window. Despite these benefits, the ability for PheWAS to substantially add to the decision making in drug development is thwarted by the difficulty to obtain and systematize comprehensive genotypes and phenotypes across very large numbers of individuals.

Here, we test the hypothesis that PheWAS can inform target validation at early stages of drug discovery. We select candidate drug targets across a range of therapeutic indications based on their support from genome-wide association studies (GWAS). To maximize power, we map a large spectrum of clinical endpoints from four of the world's largest disease-agnostic cohorts with extensive health information (23andMe, UK Biobank interim release, FINRISK and CHOP) and conduct association testing in up to 697,815 individuals. We validate the top associations in the extended UK Biobank cohort (337,199 participants), and apply conditional analyses and co-localization methods to identify true pleiotropy predicting drug efficacy or safety signals. Our results show that PheWAS, despite limitations, enrich drug discovery with valuable information.

## Results

**Assessing pleiotropy of SNPs near 19 candidate drug targets**. In this study, we queried the literature for genes nominated through GWAS as putatively causally linked to the risk for common complex human diseases and supported by various degrees of additional genetic or biological evidence. We selected 19 genes that, based on previously described genetic associations with either immune-mediated (9 genes: *ATG16L1, CARD9, CD226, CDHR3, GPR35, GPR65, IFIH1, IRF5,* and *TYK2*), cardiometabolic (8 genes: *F11, F12, GDF15, GUCY1A3, KNG1, LGALS3, PNPLA3,* and *SLC30A8*), or neurodegenerative diseases (2 genes: *LRRK2, TMEM175*), were evaluated as potential novel drug targets (Table 1). Gene-disease associations had been established through 25 common lead single nucleotide polymorphisms (SNPs) that all reached a conservative level of statistical significance ($P < 5 \times 10^{-8}$) for association in GWAS with at least one phenotype of relevance to drug discovery and development (Supplementary Table 1). All of these SNPs have either been demonstrated to impact the target gene in functional studies (genetic evidence), or locate proximal to a gene implicated in a biological mechanism related to the GWAS phenotype (biological evidence). Our selection ranged from targets with little biological knowledge beyond GWAS nomination (e.g., *TMEM175* for Parkinson's disease (PD)) to targets with drug candidates in early clinical trials (e.g., *F11* for thromboembolism). Details on the genetic and biological support for all selected genes and SNPs is provided in Supplementary Methods.

To broadly investigate pleiotropic effects of the 25 chosen SNPs in a maximal number of individuals, we interrogated four large disease-agnostic cohorts that link genome-wide genotype data from individuals of European ancestry with extensive phenotypic data: the 23andMe Inc. cohort with self-reported phenotypes on 671,151 research participants[22], the interim UK Biobank cohort analyzed by Genomics plc with questionnaire-based health information on 112,337 participants (from the first genetic data release in May 2015)[10], and two EHR-based cohorts from an adult Finnish cohort (FINRISK; 21,371 participants)[23] and from a pediatric healthcare population from the Children's Hospital of Philadelphia (CHOP; 12,044 patients)[24] (Table 2 and Methods). All four cohorts contributed phenotypic data in different formats (medical interviews, self-reports, WHO ICD codes, or ICD9-CM codes) in both shared and distinct phenotype categories (Fig. 1a). Manual phenotype mapping identified 145 distinct clinical endpoints that were tested in two or more cohorts in up to 697,815 individuals (Fig. 1b, Supplementary Table 2, and Supplementary Table 3). As illustrated in Fig. 1c, these 145 mapped phenotypes represent a broad spectrum of disease categories and, as typically observed in disease-agnostic cohorts, show significant variability in the case:control ratios, both within and between cohorts. In addition, PheWAS in the four cohorts provided association results for 1538 cohort-specific unmapped endpoints, leading to a total of 1683 endpoints included in our analysis. Association testing in the cohorts was performed using logistic regression models; meta-analyses were performed using fixed effect models (see Methods for details).

**Meta-PheWAS replicate known GWAS signals**. We first evaluated whether association testing in the four disease-agnostic cohorts replicated established results from published GWAS. GWAS had associated the 25 tested SNPs with genome-wide

**Table 1 Candidate drug targets investigated in the study**

| | Human genetics | | Drug development[b] |
|---|---|---|---|
| Gene | Prior GWAS associations[a] | Mendelian disorders (direction of effect) | Indications/status/proposed mechanism of action |
| ATG16L1 | CD; IBD | – | – / – / – |
| CARD9 | CD; IBD; UC | Familial candidiasis (LOF) | – / – / – |
| CD226 | IBD; MPV; T1D | – | – / – / – |
| CDHR3 | Asthma | – | – / – / – |
| F11 | aPTT; VTE; FXI levels | FXI deficiency (LOF) | Hemophilia C/launched/factor XI stimulant Thrombosis/phase II/factor XI inhibitor |
| F12 | aPTT; FXII levels | Hereditary angioedema (GOF); FXII deficiency (LOF) | Hereditary angioedema; thrombosis/phase I/factor XII inhibitor Antiphospholipid syndrome/preclinical/factor XII inhibitor |
| GDF15 | BMI | – | Cachexia/preclinical/GDF-15 antagonist |
| GPR35 | CD; IBD; UC | – | Cough; mastocytosis; pruritus/phase II/GPR35 agonist |
| GPR65 | CD; IBD; UC | – | – / – / – |
| GUCY1A3 | BP; CAD; MI | Moyamoya 6 with achalasia (LOF) | – / – / – |
| IFIH1 | IgAD; IBD; psoriasis; UC; SLE; T1D; vitiligo | Aicardi–Goutieres syndrome (GOF); Singleton–Merten syndrome (GOF) | Solid cancer/phase I/IFIH1 stimulant (additional targets: RIG-I; TLR3) |
| IRF5 | PBC; RA; SJO; SLE; SSc; UC | – | – / – / – |
| KNG1 | aPTT; FXI levels | – | – / – / – |
| LGALS3 | Galectin-3 levels | – | Liver fibrosis; non-alcoholic steatohepatitis; psoriasis/phase II/galectin-1 and 3 antagonist Pulmonary idiopathic fibrosis/phase II/galectin-3 antagonist Atopic eczema; head and neck cancer; melanoma/phase I/galectin-1 and 3 antagonist Arrhythmia; fibrosis: myocardial, renal; pulmonary hypertension/preclinical/galectin-1 and 3 antagonist Cardiac and renal conditions/preclinical/galectin-3 antagonist |
| LRRK2 | CD; IBD; PD; UC | Familial Parkinson's disease (GOF) | Parkinson's disease/phase I/LRRK2 inhibitor Alzheimer's disease; glaucoma/preclinical/LRRK2 inhibitor |
| PNPLA3 | Alcohol-related cirrhosis; ALT; CT; hepatic steatosis; NAFLD | – | – / – / – |
| SLC30A8 | Fasting glucose; T2D | – | – / – / – |
| TMEM175 | PD | – | – / – / – |
| TYK2 | CD; IBD; MS; PBC; psoriasis; RA; SLE; T1D; UC | Immunodeficiency (LOF) | Atopic eczema/phase II/JAK1 and TYK2 inhibitor psoriasis/phase II/TYK2 inhibitor; JAK1 and TYK2 inhibitor SLE/phase II/TYK2 inhibitor Alopecia areata; UC/phase II/JAK1 and TYK2 inhibitor IBD/phase I/TYK2 inhibitor; JAK1 and TYK2 inhibitor psoriatic arthritis/phase I/TYK2 inhibitor CD/preclinical/JAK1-3 and TYK2 inhibitor cancer: acute leukemia, colorectal, anaplastic large cell lymphoma; MS; RA/preclinical/JAK1 and TYK2 inhibitor Uveitis/preclinical/TYK2 inhibitor |

ALT: alanine aminotransferase, aPTT: activated partial thromboplastin time, BMI: body mass index, CAD: coronary artery disease, IgAD: immunoglobulin A deficiency, MI: myocardial infarction, MPV: mean platelet volume, NAFLD: non-alcoholic fatty liver disease, RA: rheumatoid arthritis, SLE: systemic lupus erythematosus, T1D: type 1 diabetes, T2D: type 2 diabetes, VTE: venous thromboembolism, CD: Crohn's disease, IBD: inflammatory bowel disease, MS: multiple sclerosis, PBC: primary biliary cirrhosis, PD: Parkinson's disease, SJO: Sjogren's syndrome, SSc: systemic sclerosis, UC: ulcerative colitis, GOF: gain-of-function, LOF: loss-of-function
[a]Published associations at the genetic locus as defined in Methods. Causal gene not always unambiguously established. For details, see Supplementary Information
[b]As listed in Citeline's Pharmaprojects database. Active development with most advanced status (preclinical or clinical) as of Dec 16, 2017 is indicated

significance to 58 binary disease endpoints. Of these, 47 endpoints were ascertained with adequate power (beta ≥ 0.8) to reach $P < 0.05$ in the PheWAS meta-analysis. After excluding the three Parkinson's disease associations that were derived from 23andMe data in the published GWAS, we observed that 33 of the 44 (75%) powered GWAS associations replicated at $P < 0.05$ in our PheWAS meta-analysis with consistent directions of effects (18/27 (67%) powered GWAS associations replicated at FDR < 0.1 ($P < 3.8 \times 10^{-4}$)) (Supplementary Figs 1, 2, and Supplementary Table 4). The overlap between the published GWAS effect sizes and the confidence intervals observed in the meta-PheWAS and in the four cohorts is provided in Supplementary Figs 2 and 3. As

expected from data obtained in real-world settings, the replication rate of known associations was highly disease-dependent (Supplementary Fig. 1B). For instance, out of the 11 associations that failed to replicate despite sufficient case numbers in the cohorts, eight were associations with inflammatory bowel disease (IBD), Crohn's disease (CD), or ulcerative colitis (UC), likely reflecting suboptimal ascertainment of these endpoints in real-world settings. Nonetheless, the high replication rate of previously reported associations demonstrates the power of combining disease-agnostic cohorts from various sources to detect and validate true SNP-disease associations, and to substantiate therapeutic hypotheses.

**Table 2 Cohorts included in this study**

| Cohort | Participants geographic distribution | Phenotypes source | N binary endpoints tested[a] | Max sample size |
|---|---|---|---|---|
| 23andMe | 89% USA (adult) | Questionnaire-based self-reports | 654 | 671,151 |
| Genomics plc UK Biobank | 100% UK (adult) | Questionnaire-based self-reports, medical interviews and follow-up | 90 | 112,337 |
| FINRISK | 100% Finns (adult) | National health registries (ICD8,9,10) | 278 | 21,371 |
| CHOP | 100% USA (pediatric) | Electronic health records (ICD9-CM) | 870 | 12,044 |
| Genomics plc GWAS | Mixed | Mixed—multiple independent disease-specific cohorts | 34 | - |

[a]Number of binary endpoints with N cases ≥ 20

**Meta-PheWAS identify novel SNP-phenotype associations**. We next investigated whether meta-PheWAS across the four cohorts could identify novel associations to support the proposed clinical indication(s) (derived from established genetic associations, see Table 1), suggest alternative indications for drug repositioning, or uncover potential target-related ADEs. To improve statistical power in this analysis, the PheWAS results in the four cohorts were meta-analyzed together with summary statistics from published GWAS studies of 34 diseases available from a larger database assembled and harmonized by Genomics plc (referred to as Genomics plc GWAS, Supplementary Note 1). Overall, 27,763 association tests (across 145 harmonized and 1538 cohort-specific endpoints) resulted in nine putative novel associations reaching study-wide significance after Bonferroni correction ($P < 1.8 \times 10^{-6}$) (Table 3). Using a less stringent significance threshold of FDR < 0.1 ($P < 7 \times 10^{-4}$) previously applied in PheWAS[25], we identified 71 distinct putative novel associations (Fig. 2, Supplementary Table 5 and Supplementary Data 1). Of these, 30 were with mapped phenotypes and were obtained from meta-analyzing results from at least two cohorts, and 41 were supported by a single cohort (and thus require independent replication) (Supplementary Table 5). Forty-three of these putative novel associations showed the same directions of effect as disease endpoints related to the proposed clinical indication for a drug and may hint at potential repositioning opportunities (Supplementary Fig. 4). Conversely, 27 showed directions of effect opposite to disease endpoints related to the proposed clinical indication and may suggest safety signals that could endanger therapeutic success and warrant monitoring for in preclinical models and clinical trials (Supplementary Fig. 4).

The 30 novel associations with mapped phenotypes showed limited evidence of heterogeneity between the PheWAS cohorts (Supplementary Fig. 5). Twenty-three (77%) of these 30 associations showed an $I^2 < 40\%$. Manual review of the results showed that only one of the seven associations with $I^2 > 40\%$, the GDF15 rs17724992 association with high blood pressure, was less significant in the meta-analysis than in the individual cohorts ($P_{23andMe} = 6.4 \times 10^{-10}$, $OR_{23andMe} = 0.96$; $P_{Gplc/UK\ Biobank} = 0.58$, $OR_{Gplc/UK\ Biobank} = 0.99$; $P_{meta} = 7.6 \times 10^{-9}$, $OR_{meta} = 0.97$) (Supplementary Fig. 5B).

**Replication of novel associations in UK Biobank v2**. Forty-one of the 71 potential novel associations reaching FDR < 0.1, including eight of the nine novel associations reaching study-wide significance, were with phenotypes tested by Neale et al. through GWAS in the expanded UK Biobank (v2) cohort of up to 337,199 participants of European ancestry. In an attempt to replicate putative novel associations discovered in our meta-PheWAS, we performed weighted Z score-based meta-analyses between the 23andMe, FINRISK and CHOP PheWAS results, the published GWAS results and the UK Biobank v2 results (excluding the Gplc

UK Biobank results). Out of the 41 putative novel associations, 16 showed P < 0.05 in UK Biobank v2 with consistent direction of effect, thus validating and further strengthening significance of our previous results (Supplementary Table 6). An additional seven potential novel associations showed increased significance in meta-analysis despite P > 0.05 in UK Biobank v2, largely due to small number of cases and lack of statistical power in UK Biobank v2 alone. Overall, meta-analysis with UK Biobank v2 strengthened all eight novel associations with study-wide significance after Bonferroni correction and 23/41 (56%) of the potential novel associations with FDR < 0.1, including eight associations that were based on results from a single PheWAS cohort. Strengthened associations in the meta-analysis with UK Biobank v2 include the rs17724992-high blood pressure association that showed significant heterogeneity between the 23andMe and the interim UK Biobank cohorts ($P_{23andMe} = 6.4 \times 10^{-10}$, $OR_{23andMe} = 0.96$; $P_{UK\ Biobank\ v2} = 4.4 \times 10^{-5}$; $P_{meta\_v2} = 3.9 \times 10^{-13}$).

**Interpretation of apparent pleiotropy in PheWAS results**. A challenge to the PheWAS approach is to reliably distinguish true pleiotropic associations of a SNP (or SNPs in strong LD with the lead SNP), suggesting a shared causal mechanism, from unrelated associations driven by independent SNPs at a locus[18]. For instance, in our study, the putative association of rs2274273 near LGALS3 (encoding the galactin-3 protein) with PD ($OR_{23andMe} = 0.94$, $P_{23andMe} = 1 \times 10^{-4}$) likely reflects a distinct causal mechanism previously attributed to GCH1[26]. rs2274273 is a protein quantitative trait locus (pQTL) that controls plasma levels of galectin-3[27]. Through a Bayesian test for co-localization using summary statistics from published GWAS studies[26,28,29], we excluded rs2274273 as a causal SNP for PD (posterior probability for a shared variant leading the PD and galectin-3 levels associations = 0.0008%) (Supplementary Fig. 6).

A second challenge to PheWAS is the existence of common co-morbidities among endpoints, or alternatively an insufficient distinction between phenotypes[19]. In our meta-PheWAS, rs17724992 near GDF15 showed association with multiple cardiovascular-related phenotypes, which is likely mediated by the known association of this SNP with body mass index (BMI)[30], an established risk factor for cardiovascular disease[31]. This is supported by the lack of association of rs17724992 with blood pressure ($P_{SBP} = 0.064$, $P_{DBP} = 0.134$) and coronary artery disease (CAD, $P = 0.17$) in the large GWASs published by the International Consortium for Blood Pressure and the CARDIoGRAMplusC4D consortium[32,33]. Phenotype correlation scores can indicate apparent pleiotropic effects that may be explained by comorbidities or confounding (Supplementary Fig. 7), yet follow-up customized association analyses adjusting for specific phenotypic covariates are required to distinguish true pleiotropic effects and inform target validation.

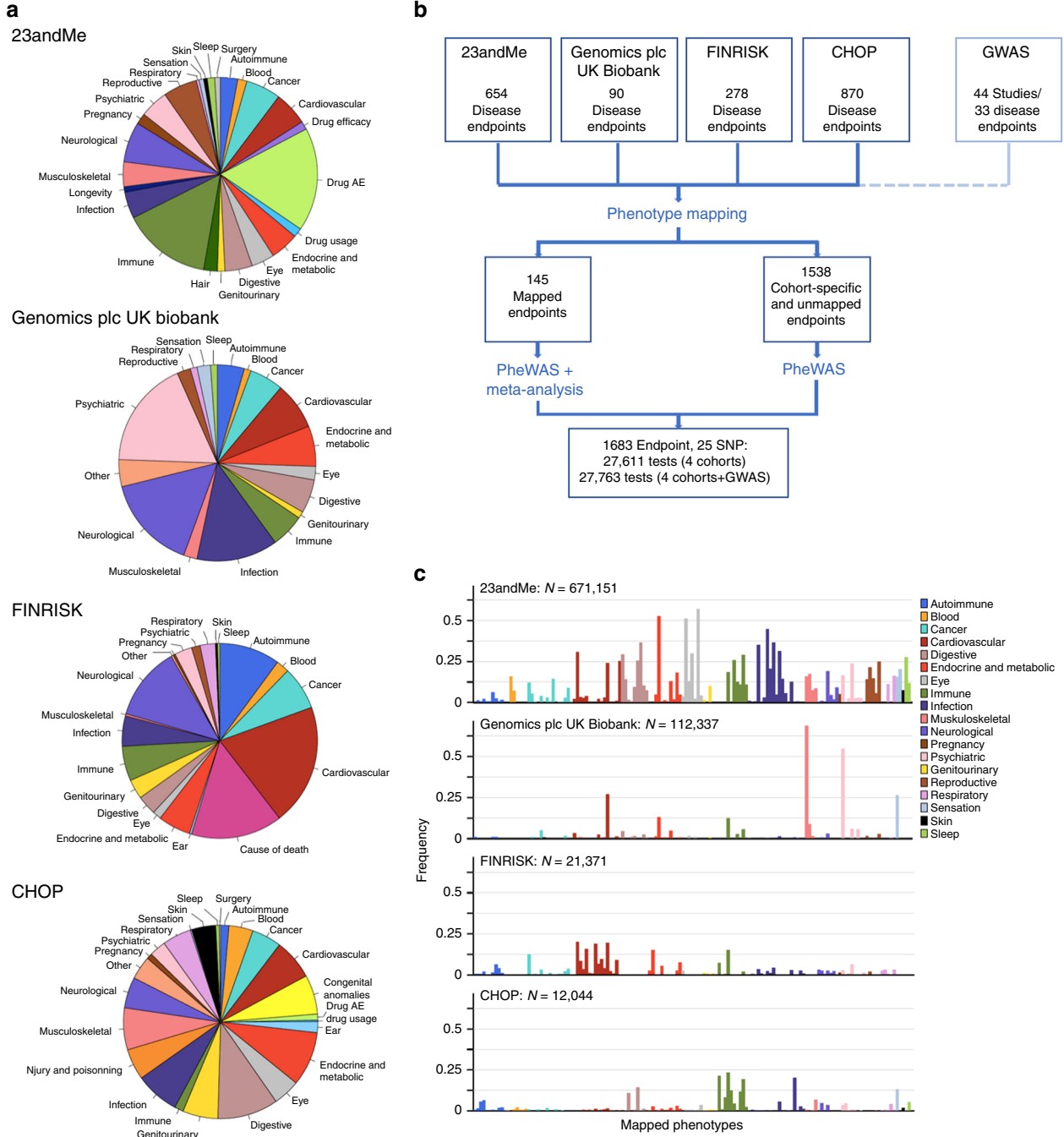

**Fig. 1** Phenotypes tested and study design. **a** Categories of phenotypes assessed in the 23andMe, Genomics plc UK Biobank, FINRISK, and CHOP cohorts. **b** Manual phenotype mapping was performed to identify phenotypes shared between cohorts. One hundred and forty-five phenotypes were captured with at least 20 cases in at least 2 cohorts. After PheWAS in each cohort separately, the 145 phenotypes were meta-analyzed to increase statistical power and enable systematic comparisons of results between cohorts. **c** The 145 mapped phenotypes (see Supplementary Table 2) represent a broad spectrum of phenotypic categories and are captured with variable case:control ratios in the cohorts tested

In summary, these two examples demonstrate that thorough investigation of association results can reduce biases introduced through PheWAS.

**Meta-PheWAS reveal pleiotropic effects of *PNPLA3* rs738409.** Among the nine study-wide significant associations, our meta-PheWAS revealed multiple novel associations for the *PNPLA3* missense SNP rs738409 (p.I148M). The rs738409-G allele has previously been reported as associated with an increased risk for non-alcoholic fatty liver disease (NAFLD), alcohol-related cirrhosis and hepatic steatosis, as well as elevated alanine aminotransferase (ALT) levels, most likely through a gain-of-function (GOF) mechanism (Supplementary Methods). Consistent with these findings, our meta-PheWAS found rs738409-G to be associated with elevated liver tests (OR$_{23andMe}$ = 1.25, $P_{23andMe}$ = $4 \times 10^{-45}$) (Supplementary Fig. 8). Beyond that, our analysis also indicated that carriers of the rs738409-G allele that increases ALT are more prone to develop liver toxicities when treated with nonsteroidal anti-inflammatory drugs (NSAIDs) such as ibuprofen (OR$_{23andMe}$ = 1.43, $P_{23andMe}$ = $4.6 \times 10^{-5}$) or aspirin

**Table 3 Significant novel associations in the PheWAS meta-analysis**

| Gene | SNP | EA (EAF)[b] | Known associated phenotype[c] | Novel association in meta-PheWAS[a] | | | | | |
|------|-----|-------------|-------------------------------|-------------------------------------|---|---|---|---|---|
| | | | | Phenotype | OR (CI95) | P value | Direction[d] | N cases | N controls |
| CD226 | rs763361 | T (0.47) | IBD | Hypothyroidism | 1.05 (1.04-1.07) | 8.11e−11 | ++?+? | 35,428 | 412,577 |
| GDF15 | rs17724992 | A (0.73) | BMI | Heart metabolic disease[e] | 1.03 (1.02-1.04) | 3.08e−09 | +???? | 275,944 | 209,302 |
| | | | | High blood pressure[e] | 1.03 (1.02-1.04) | 7.64e−09 | ++??? | 151,511 | 465,686 |
| | | | | Blood pressure medication[e] | 1.03 (1.02-1.04) | 1.76e−07 | +???? | 125,406 | 394,753 |
| | | | | GERD | 1.03 (1.02-1.04) | 6.11e−07 | +???? | 130,654 | 384,572 |
| | | | | Any CVD[e] | 1.03 (1.01-1.04) | 1.40e−06 | +???? | 148,577 | 388,405 |
| IFIH1 | rs1990760 | T (0.61) | T1D | **Asthma[f]** | **0.96 (0.95-0.98)** | **1.11e−07** | − − − − − | 57,101 | 269,659 |
| IRF5 | rs10488631 | C (0.11) | SLE | Hypothyroidism | 1.08 (1.05-1.12) | 5.78e−07 | ++?+? | 23,182 | 236,240 |
| PNPLA3 | rs738409 | G (0.33) | ALT | **Severe acne** | **0.91 (0.88-0.93)** | **1.47e−11** | −???? | 14,812 | 187,018 |
| | | | | **High cholesterol** | **0.96 (0.94-0.97)** | **1.59e−07** | − −??? | 101,646 | 180,947 |
| TYK2 | rs34536443 | G (0.89) | Psoriasis | Any immune disorder | 1.10 (1.07-1.13) | 4.27e−12 | +???? | 112,148 | 173,986 |
| | | | | Hypothyroidism | 1.14 (1.08-1.20) | 1.19e−06 | ++?−? | 23,145 | 233,757 |

ALT: alanine aminotransferase, BMI: body mass index, CVD: cardiovascular disease, EA: effect allele, EAF: effect allele frequency, GERD: gastroesophageal reflux disease, IBD: inflammatory bowel disease, SLE: systemic lupus erythematosus, T1D: type diabetes, T2D: type 2 diabetes

[a]Associations reaching $P < 1.8e−6$ (Bonferroni-corrected significance threshold) in the meta-analysis of PheWAS results with GWAS results. The full list of potential novel SNP-phenotype pairs reaching FDR < 0.1 is provided in Supplementary Table 5. Novel associations with direction of effect opposite to the known associated disease(s) effect, predicting potential adverse drug events, are highlighted in bold

[b]The effect allele is the risk allele for known associated disease(s) related to the therapeutic hypothesis

[c]Known associated disease related to the therapeutic hypothesis (surrogate for efficacy). The strongest association reported in the literature is indicated. The full list of known associations is provided in Supplementary Table 1

[d]Direction of effect in 23andMe, Genomics plc UK Biobank, FINRISK, CHOP, and GWAS

[e]Correlated phenotypes

[f]Meta-analysis results including the 23andMe, Gplc/UK Biobank, FINRISK, CHOP, and GWAS Gabriel cohorts. When further including the independent GWAS EVE study, the association reaches $P = 6.7 \times 10^{-8}$

($OR_{23andMe} = 1.57$, $P_{23andMe} = 5.3 \times 10^{-5}$). It also confirmed the association of rs738409-G with increased risk of T2D ($OR_{meta} = 1.08$, $P_{meta} = 8 \times 10^{-11}$) recently reported in a T2D fine-mapping study that confirmed rs738409 as the most likely causal SNP[34]. Our meta-PheWAS further revealed significant associations between rs738409-G and decreased risk for high cholesterol ($OR_{meta} = 0.96$, $P_{meta} = 1.6 \times 10^{-7}$; $P_{meta\_v2} = 1.1 \times 10^{-8}$) and the intake of cholesterol-lowering medications ($OR_{23andMe} = 0.97$, $P_{23andMe} = 2 \times 10^{-4}$; $P_{meta\_v2} = 2.8 \times 10^{-5}$), consistent with recent results from the lipids exome chip study describing a significant association of rs738409-G with decreased LDL levels[35]. In addition, the meta-PheWAS revealed novel significant associations between the rs738409-G GOF allele and decreased risk for acne ($OR_{23andMe} = 0.90$, $P_{23andMe} = 1.5 \times 10^{-11}$; $P_{meta\_v2} = 7.3 \times 10^{-12}$), gout ($OR_{meta} = 0.92$, $P_{meta} = 4.1 \times 10^{-5}$; $P_{meta\_v2} = 3.9 \times 10^{-9}$), and gallstones ($OR_{meta} = 0.95$, $P_{meta} = 2.7 \times 10^{-4}$; $P_{meta\_v2} = 1.5 \times 10^{-5}$). All these associations remained prominent after adjusting for elevated liver tests (Supplementary Table 7), and were further strengthened in the meta-analysis with the expanded UK Biobank cohort (Supplementary Table 6). Taken together, our PheWAS results support the hypothesis that therapeutic inhibition of PNPLA3 could treat liver diseases. They also support T2D as a potential alternative indication for PNPLA3 inhibition. However, concomitant inverse associations with multiple other endpoints, including acne and high plasma cholesterol levels, indicate potential clinically relevant on-target ADEs that should be considered in decisions to progress PNPLA3 inhibitors toward clinical development.

**IFIH1 partial loss-of-function increases asthma risk.** The meta-PheWAS further revealed novel, important pleiotropic effects for drugs directed toward *IFIH1*. Carriers of the *IFIH1* (encoding MDA5) rs1990760-C allele (MAF = 40%) have an established lower risk for several autoimmune diseases (type 1 diabetes, T1D; vitiligo; systemic lupus erythematosus, SLE; psoriasis) and an increased risk for UC (Supplementary Methods). Functional studies suggest that rs1990760-C (p.T946A) causes *IFIH1* loss-of-function (LOF), and additional *IFIH1* LOF alleles have been shown to protect against T1D, vitiligo, psoriasis, and psoriatic arthritis (PsA) (Supplementary Methods). Our meta-PheWAS support these associations (Fig. 2 and Supplementary Table 4). Beyond this, we found a significant novel association between

rs1990760-C and increased risk for asthma ($OR_{meta} = 1.04$, $P_{meta} = 6.7 \times 10^{-8}$) that reached $P_{meta\_v2} = 2 \times 10^{-8}$ in the meta-analysis with the expanded UK Biobank cohort (Fig. 3a and Supplementary Table 6). The association between rs1990760 and asthma was supported by data from all four disease-agnostic cohorts as well as the GABRIEL and EVE asthma GWAS cohorts[36,37], despite lack of power to detect an association with rs1990760 in the published GWAS cohorts alone (Fig. 3b). This association remained significant after adjustment for auto-immune diseases in the 23andMe cohort, demonstrating that the asthma association is independent of the previously established associations of rs1990760 with autoimmunity (Supplementary Table 8). Co-localization analysis confirmed that the same SNP was responsible for the SLE, UC, and asthma associations at the locus, supporting true pleiotropic effects driven by the same causal variant(s) (Fig. 3c). The observed *IFIH1* pleiotropic effects were further strengthened by the observation in the Genomics plc UK Biobank data that the independent low-frequency *IFIH1* missense allele p.I923V (rs35667974-C, MAF = 1.8%), previously reported to result in *IFIH1* LOF and to protect against T1D, vitiligo, psoriasis, and PsA, and to increase risk of UC, was also associated with increased risk of asthma ($OR_{Gplc/UK \ Biobank} = 1.18$, $P_{Gplc/UK \ Biobank} = 1.1 \times 10^{-4}$) (Fig. 3d). Together, these and previous findings establish *IFIH1* as a gene with an allelic series[38] and further support the therapeutic hypothesis that inhibition of MDA5 may protect against several autoimmune diseases. However, our results also reveal the potential of MDA5 inhibitors to cause pulmonary ADEs and strengthen previous findings for an increased risk for colitis-related symptoms, endpoints that may limit the therapeutic window of MDA5 modulators and should be considered for monitoring in clinical trials.

**PheWAS assist target prioritization for thromboembolism.** Beyond informing on individual genes, we hypothesized that PheWAS might help prioritize targets among several candidates within a biological pathway. Factors XI, XII, and plasma kini-nogen (encoded by *KNG1*) are members of the contact activation coagulation pathway[39]. Anti-coagulation therapies directed against these factors are hypothesized to have improved therapeutic windows over current standard-of-care, which is accompanied by significant bleeding liabilities[40]. With the aim to estimate genetic risk–benefit profiles for the three candidate

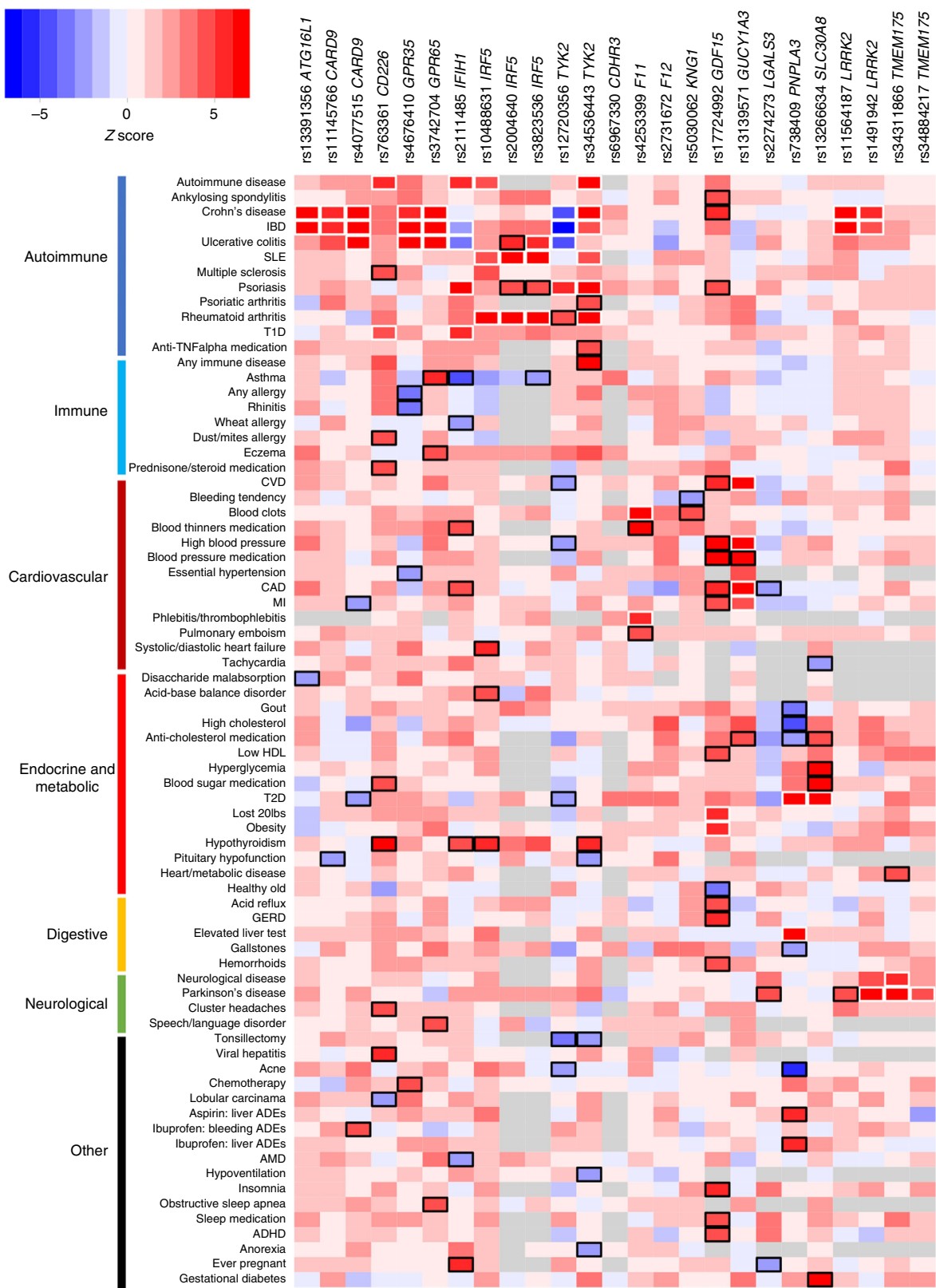

**Fig. 2** Meta-PheWAS results for 25 SNPs in candidate drug targets. Phenotypes associated at FDR < 0.1 (P < 7e−4) with at least one SNP in the meta-PheWAS are represented. Direction of effect of the known disease-risk increasing allele related to the therapeutic hypothesis is indicated. A positive Z score (in red) indicates increased risk, a negative Z score (in blue) indicates reduced risk. Known and novel associations reaching FDR < 0.1 are outlined in white and black respectively. Detailed association results are provided in the Supplementary Data 1

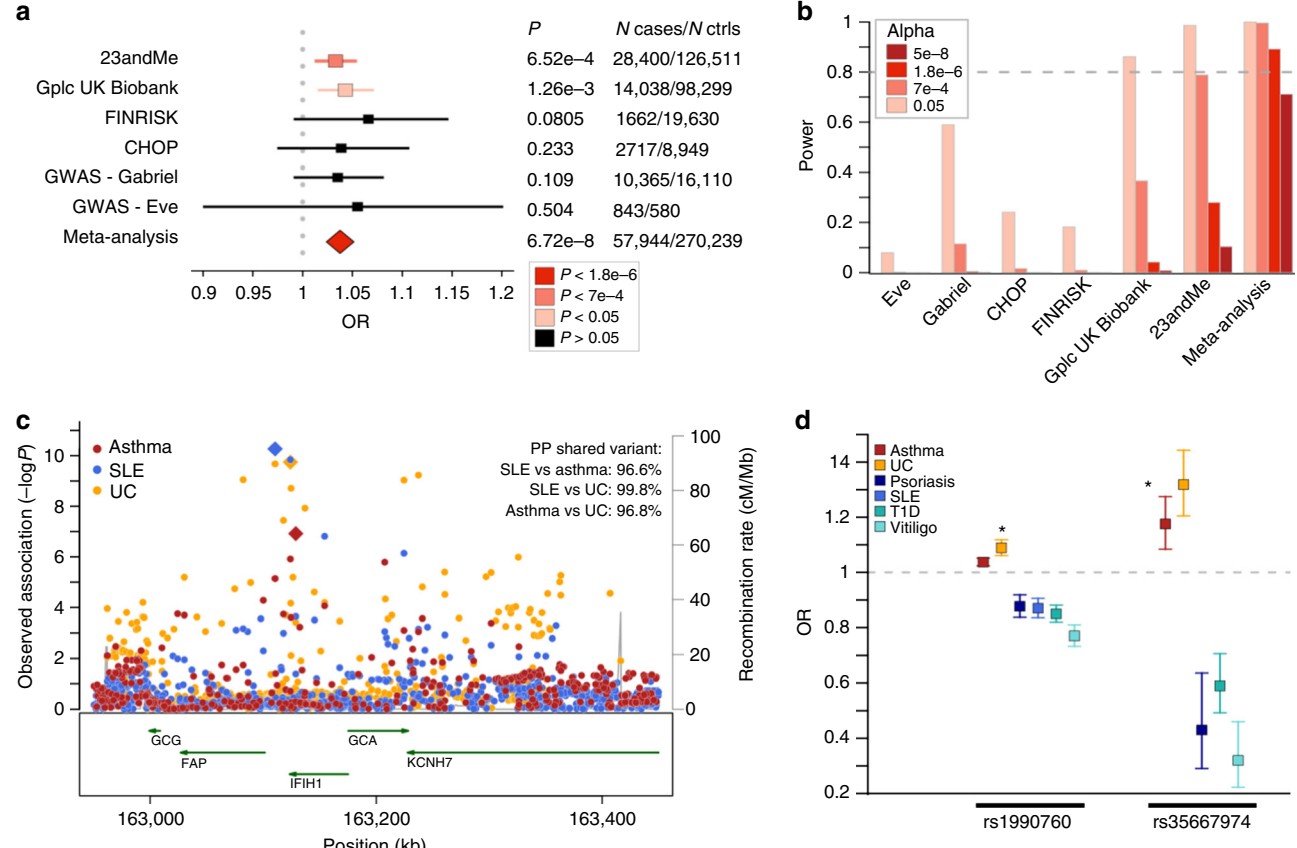

**Fig. 3** Pleiotropic effects of *IFIH1* LOF variants. **a** A significant association of *IFIH1* rs1990760-C (p.T946A) with increased risk of asthma was observed in the meta-analysis of PheWAS and GWAS results, with consistent effect estimate across the six cohorts tested. Odds ratios (OR) and 95% confidence intervals are represented. **b** Power estimation demonstrates the lack of power to detect an association at rs1990760-C in currently available asthma GWAS studies. Power to surpass various significance cutoffs ($P < 0.05$; FDR < 0.1, $P < 7e-4$; study-wide significance after Bonferroni correction, $P < 1.8e-6$; and genome-wide significance, $P < 5e-8$) in the six cohorts was estimated using the frequency of the asthma risk allele (RAF = 0.39), the OR in the PheWAS/GWAS meta-analysis (OR = 1.037), a disease prevalence of 8%, and the number of cases and controls in each of the cohorts. **c** Co-localization analysis demonstrates that the asthma, systemic lupus erythematosus (SLE), and ulcerative colitis (UC) associations at the *IFIH1* locus are driven by a shared causal signal. Regional association results with asthma (red), SLE (blue) and UC (orange) are shown. PP, posterior probability of co-localization. **d** Results from this study (indicated by an asterix) combined with previously published findings suggest an allelic series of LOF *IFIH1* alleles decreasing the risk of various autoimmune diseases while increasing the risk of asthma and UC. OR and 95% confidence intervals of association for the *IFIH1* loss-of-function alleles rs1990760-C (p.T946A) and rs35667974-C (p. I923V) are shown

targets, we chose to interrogate three uncorrelated SNPs at the *F11*, *KNG1*, and *F12* loci. These three SNPs had similar allele frequencies in Europeans, had previously been shown to impact FXI, FXII, and/or KNG1 mRNA and/or protein levels, and are associated with activated partial thromboplastin time (aPTT), a biomarker of blood clotting, or venous thromboembolism (VTE) risk (Supplementary Methods and Supplementary Table 1). Carriers of the rs4253399-T allele, which reduces circulating FXI levels and increases aPTT, showed an expected lower risk for blood clots (OR$_{meta}$ = 0.84, $P_{meta}$ = 3.5 × 10$^{-25}$)[41], but no evidence for association with bleeding tendency (OR$_{23andMe}$ = 1.04, $P_{23andMe}$ = 0.35) (Fig. 4). In contrast, carriers of the *KNG1* allele rs5030062-A, which reduces plasma kininogen as well as circulating FXI, and increases aPTT, showed both reduced blood clotting (OR$_{meta}$ = 0.93, $P_{meta}$ = 1.6 × 10$^{-4}$) as well as increased bleeding liability (OR$_{23andMe}$ = 1.14, $P_{23andMe}$ = 4.1 × 10$^{-4}$). A nominal association with both phenotypes was found in carriers of the FXII levels-reducing and aPTT-increasing allele rs2731672-T (blood clots: OR$_{23andMe}$ = 0.96, $P_{23andMe}$ = 0.034; bleeding tendency: OR$_{23andMe}$ = 1.09, $P_{23andMe}$ = 0.039).

By comparing these results with the effect of the three SNPs on aPTT (Supplementary Table 1), our study suggests that, among the three factors tested, targeting FXI may yield the best compromise between thromboembolism risk reduction and increased bleeding liability, which is consistent with the outcomes of a recent phase 2 clinical trial[42].

## Discussion

Our study investigates the utility of PheWAS to help predict therapeutic success of candidate drug targets nominated through human genetics. We focused on a selection of loci that GWAS have firmly established as associated with common immune-mediated, cardiometabolic, or neurodegenerative human diseases, and where additional biological or genetic evidence supports candidate drug target genes within these loci as likely causing the disease associations. We analyzed SNPs impacting these targets for association with 1683 disease endpoints captured in four large, disease-agnostic population cohorts that link genome-wide genotypes with various types of structured health information. Our PheWAS meta-analysis replicates 75% of the published GWAS associations at $P < 0.05$, substantially surpassing performance of previous PheWAS in smaller cohorts[25]. Through meta-analyzing PheWAS results with published GWAS data, we identified nine novel SNP-phenotype

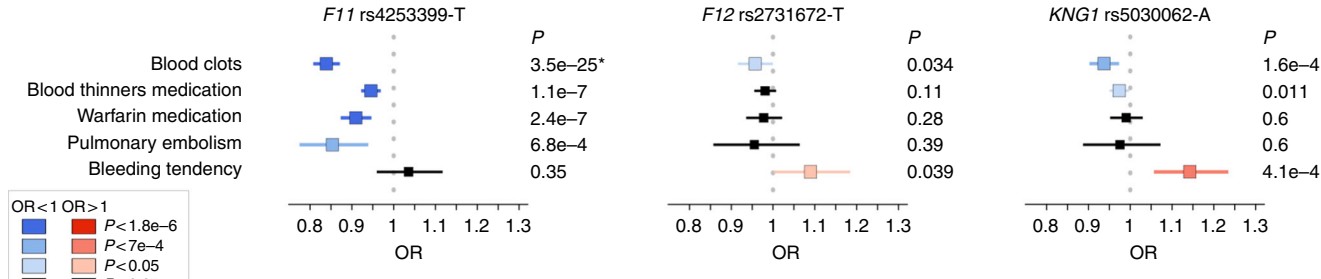

**Fig. 4** PheWAS for contact activation coagulation pathway targets. Three SNPs known to affect plasma protein levels of FXI (rs4253399), FXII (rs2731672), and KNG1 (rs5030062), and previously reported associated with partial thromboplastin time (aPTT) were interrogated in meta-PheWAS. Five phenotypes were observed as significantly associated (FDR < 0.1) with at least one of the three SNPs: blood clots (23andMe, FINRISK, and CHOP: 7487 cases, 273,305 controls), known association with the *F11* SNP (*), blood thinners medication (23andMe: 22,985 cases, 236,431 controls), warfarin medication (23andMe: 7142 cases, 94,701 controls), pulmonary embolism (Gplc/UK Biobank: 949 cases, 111,077 controls), and bleeding tendency (23andMe: 1574 cases, 85,223 controls). Odds ratios (OR) and 95% confidence intervals of association of the aPTT-increasing alleles are shown. Detailed association results are provided in the Supplementary Data 1

associations that exceeded stringent significance thresholds for multiple test correction, as well as additional putative associations with therapeutically relevant clinical endpoints. For a subset of early drug targets, our results support previous genetic evidence for efficacy in distinct common disease indications. Our analysis further proposes alternative indications as opportunities for drug repositioning and predicts on-target adverse drug events that may warrant preclinical or clinical monitoring.

Among others, we discovered novel associations for p.I148M in *PNPLA3*. This is a common gain-of-function missense allele increasing the risk for a range of liver phenotypes, which suggested that pharmaceutical inhibition of PNPLA3 could be a viable strategy to treat or prevent liver diseases. While our PheWAS support this hypothesis and further backs expanding the indication spectrum of a putative PNPLA3 inhibitor to T2D, we also uncovered opposite associations with severe acne and high cholesterol, phenotypes that if observed during a clinical trial might put a therapeutic program at risk.

We also identified a novel association of the *IFIH1* loss-of-function allele rs1990760-C (p.T946A) with risk of asthma. The rs1990760-C allele, which protects against several autoimmune diseases and increases risk of UC, has been shown to decrease interferon (IFN) signaling and lower resistance to viral challenge[43], while complete loss of IFIH1 function makes children susceptible to severe viral respiratory infections[44,45]. The association of rs1990760-C with increased risk of asthma discovered in our meta-PheWAS is consistent with the observation that bronchial epithelial cells from asthmatics produce lower amounts of IFN-β during viral infections[46], a finding that lead to inhaled IFN-β being tested in phase 2 clinical trials for the treatment of virus-induced asthma exacerbation[47]. Future studies will need to investigate the risk:benefit ratio of modulating MDA5 (encoded by *IFIH1*) for asthma relative to autoimmune diseases.

While our study illustrates the power of systematically interrogating disease-agnostic cohorts with extensive health information to enrich target validation, it also emphasizes several opportunities to improve existing resources in order for PheWAS to become a routine tool in drug discovery and development. First, truly large, thoroughly phenotyped cohorts will be needed to adequately power PheWAS. Despite our meta-PheWAS being conducted in close to 700,000 individuals, 20% of GWAS associations could not be replicated (P < 0.05) in the disease-agnostic cohorts due to an insufficient number of cases. In addition, PheWAS should considerably gain from improved phenotypic endpoints[48]. In our study, this is best reflected by an only modest replication rate, despite adequate power, for CD, UC, and IBD

endpoints that are closely related and difficult to discern from other disorders in routine clinical settings[49]. To better take these considerations and other characteristics of disease-agnostic cohorts (typical case:control ratio unbalance between phenotypes and phenotype correlation) into account, novel statistical methods will be needed to better define significance thresholds and control type I error rates in PheWAS[50]. Second, our study highlights the challenge to systematically combine phenotypes from independent disease-agnostic cohorts with various phenotype data sources. While we introduce the concept of meta-PheWAS and demonstrate that mapping phenotypes to interrogate independent PheWAS cohorts may considerably strengthen association signals, there is still a need for standardized terminology, automated phenotype extraction, and coordinated data management across healthcare institutions that will help with better harmonization across cohorts in the future[9,51]. A third challenge to the PheWAS approach is inherent to the current limitations of human genetics. Even when starting from a highly-annotated set of loci as in our study, PheWAS may lead to spurious interpretation of association results that can only be ruled out through thorough follow-up[18]. We demonstrate this at the example of *LGALS3* and PD. Access to genome-wide association results for systematic fine-mapping and co-localization analyses, functionalization of GWAS loci and the emergence of association data for intermediate phenotypes, e.g., at the protein level, will be needed to help narrow the gap between SNPs and candidate target genes in the future. Finally, a fourth challenge to broadly use PheWAS for drug development is to relate findings from germline variants that impact a target across an individual's entire lifetime to success of an interventional trial with much shorter observation periods. In the end, many decisions to pursue or discontinue a therapeutic program may remain dependent on the specific risk:benefit ratio that quantitative genetics as applied here may help to predict, and the level of unmet clinical need.

Taken together, our study highlights PheWAS as a highly promising, yet largely untapped opportunity to use disease-agnostic cohorts with extensive health information for drug target validation. We provide several examples that illustrate PheWAS as a powerful strategy to help predict efficacy and unintended drug effects, which should ultimately help to develop better drugs. Whether PheWAS may truly impact decision making during drug development will only become evident with either the emergence of ADEs in trials that genetics could have predicted, or reduced safety-related attrition rates for portfolios enriched in targets nominated through human genetics. The growing number of large-scale population cohorts that link genetic data with

extensive health data, together with an increased willingness across the borders of academia, biotech and the pharmaceutical industry to collaborate and share data, will provide opportunities to demonstrate that.

## Methods

**SNP selection.** In this study, we selected 25 SNPs that were significantly associated ($P < 5 \times 10^{-8}$) in published GWAS with binary or quantitative phenotypes related to three main therapeutic areas: (auto)immune, cardiometabolic, or neurodegenerative diseases (Supplementary Methods). These 25 SNPs had either been functionally validated in published studies, establishing the candidate target gene as causal for the risk of disease, or they were located within or near genes (as defined by the regions encompassing all SNPs in $r^2 > 0.5$ to the GWAS index SNPs extended to the nearest recombination hot spots) for which previous studies had generated convincing biological evidence to be of relevance for the respective clinical endpoint. The 25 SNPs were linked to 19 genes that were evaluated as candidate drug targets. Detailed information on the SNPs, candidate causal genes and their link to common human disease is provided in Supplementary Methods. The list of SNPs and their known associated phenotypes is provided in Supplementary Table 1.

**Study cohorts.** We interrogated four large observational disease-agnostic cohorts of subjects of European ancestry with genome-wide genotyped data linked to extensive phenotypic information (Table 2). All participants included in each of the four cohorts were unrelated individuals of European ancestry. Individual-level data from each cohort was analyzed independently, and the relevant summary statistics for the 25 SNPs were shared for further analysis. We restricted all cohorts to binary disease phenotypes with at least 20 cases per cohort. All endpoints were derived from questionnaires or ICD codes (including endpoints like high cholesterol or high blood pressure). No quantitative laboratory measurements were included in the study.

The 23andMe cohort comprised up to 671,151 participants and 654 binary disease endpoints derived from questionnaire-based self-reports[22]. Participants were restricted to a set of individuals who have > 97% European ancestry, as determined through an analysis of local ancestry using a support vector machine (SVM) and a hidden Markov model (HMM) to assign individuals to one of 31 reference populations. For each phenotype, we chose a maximal set of unrelated individuals using a segmental identity-by-descent (IBD) estimation algorithm. We defined individuals as related if they shared > 700 cM IBD on either one or both of their chromosomes. SNPs with Hardy–Weinberg equilibrium $P < 10^{-20}$, call rate < 95%, or strong allele frequency deviation from European 1000 Genomes reference data were excluded. Participant genotype data were then imputed against the September 2013 release of 1000 Genomes Phase1 reference haplotypes[52], using an internally developed phasing tool, Finch, which implements the Beagle haplotype graph-based phasing algorithm[53], and Minimac2[54].

The Genomics plc analysis of UK Biobank cohort (referred to as 'Genomics plc UK Biobank') comprised 112,337 participants and 90 binary disease endpoints derived from questionnaire-based self-reports and medical interviews[10]. GWAS analyses were performed by Genomics plc using the interim data release (May 2015). QC followed the recommendations provided by UK Biobank. European ethnicity was defined as self-reported white British ethnic background, and confirmed by principal component analysis clustering. Samples with relatives (3rd degree or closer) were excluded. Imputation was carried out by the UK Biobank data providers using SHAPEIT3[55], IMPUTE3[56], and a reference panel combining the 1000 Genomes Phase 3[57] and UK10K datasets[58].

FINRISK is a collection of cross-sectional population surveys carried out since 1972 to assess the risk factors of chronic diseases and health behavior in the working age population of Finland[23]. The FINRISK cohort comprised 21,371 Finnish participants and 269 binary disease endpoints derived from ICD codes grouping in Finnish national hospital registries and cause-of-death registry, and drug reimbursement and purchase registries. The FINRISK samples were genotyped using Illumina CoreExome, OmniExpress, and 610K chips. After gender check, samples with genotype missing rate > 5% or excess heterozygosity (> 4SD) were excluded. SNPs QC, including exclusion of SNPs with genotype missing > 2%, minor allele frequency <1%, or Hardy–Weinberg equilibrium $P$ value $<1 \times 10^{-6}$, was performed for each genotyping chip separately. Multidimensional scaling (MDS) components were estimated with PLINK v1.9[59] from the LD-pruned genotype data where relatives with pi-hat > 0.2 had been removed. Samples with non-Finnish ancestry observed as MDS outliers were removed. Imputation was performed with SHAPEIT[55] and IMPUTE2[56] using a reference panel combining information from the 1000 Genomes phase 3[57] and 1941 Finnish SiSu whole genome sequences[60]. Imputation was stratified based on genotyping chip.

The cohort from the Children's hospital of Philadelphia (CHOP) comprised 12,044 pediatric patients and 870 binary disease endpoints derived from ICD9–CM codes using the ICD9-to-PheWAS codes mapping described by Denny et al.[24,61]. Subjects included in the CHOP PheWAS were genotyped on one of the following genotyping chips following the Illumina standard protocols: Illumina Human610-Quad version 1, Illumina 550K SNP array, or Illumina OmniExpress array. Samples with genotype call rate > 95% were included in the study. SNPs with genotype missing rate > 5%, minor allele frequency <1%, and Hardy–Weinberg

equilibrium $P$ value < 0.00001 were excluded. Principle component analysis (PCA) was performed using EIGENSTRAT[62] on ~130,000 SNPs that had been pruned for linkage disequilibrium using PLINK v1.07[59] and reference genotypes from the HapMap consortium[63]. Imputation was performed with SHAPEIT v2[55] and IMPUTE2[56] using the 1000 Genomes project phase 1 reference panel[52]. SNPs with INFO scores < 0.9 were excluded.

All the participants in the 23andMe, Genomics plc UK Biobank, FINRISK, and CHOP cohorts provided written informed consent for participating in research studies. Blood or saliva samples were collected according to protocols approved by local institutional review boards. Details are provided in the original publications describing the cohorts[10,22–24]. This research has been conducted using the UK Biobank resource under the Genomics plc project application number 9659.

In addition, with the aim to replicate novel associations identified in the four disease-agnostic cohorts, we interrogated genome-wide summary statistics from 57 published GWAS, including 34 binary disease phenotypes, derived from a larger database that has been assembled and harmonized by Genomics plc (referred to as 'Genomics plc GWAS'). The full list of studies in Genomics plc GWAS database and tested in this study is available in the Supplementary Note 1). These included checks to ensure consistency of the data, and alignment of alleles to the forward strand of the human reference sequence, with effects ascribed to the alternative allele. Effect size estimates for quantitative traits were rescaled relative to the residual variance. Summary-statistic imputation was applied to infer association evidence at common variants (minor allele frequency > 2%) in the 1000 Genomes EUR reference panel. Results for SNPs associated with the relevant phenotype with $P < 0.05$ were included in the meta-analysis.

Correlation between all GWAS was estimated to ensure that no GWAS included in the meta-analysis for a given phenotype presented overlapping samples. In addition, to further prevent GWAS results from overlapping samples to be meta-analyzed, only the most recent/largest study for a given disease was included in our analysis when several GWAS studies in the Genomics plc database investigated the same disease. Although we could not directly estimate potential overlapping samples between the different disease-agnostic cohorts, significant overlap is very unlikely based on the participants' characteristics (Table 2).

**Identification of shared phenotypes.** The phenotypic endpoints tested in the 23andMe, UK Biobank, FINRISK, and CHOP cohorts were derived from different sources (self-reports, self-reports and medical interviews, WHO ICD codes, and ICD9-CM codes, respectively) and named using different nomenclatures (e.g., clinical terms versus popular terms, abbreviations versus full names). In order to compare and combine results from the four cohorts with published GWAS results from the Genomics plc database, we manually mapped the phenotypes. Examples of mapped and unmapped phenotypic endpoints are provided in Supplementary Table 2. This step allowed us to identify 145 distinct phenotypes shared by at least 2 cohorts and with at least 20 cases in the independent cohorts (Fig. 1). The full list of mapped phenotypes is provided in Supplementary Table 3 and the Supplementary Data 1. We note that, in each cohort some phenotypes were captured multiple times by different endpoints with slightly different definitions. In this case, only one endpoint per cohort was selected for meta-analysis.

**PheWAS and meta-analysis.** Phenome-wide association analyses for each of the 25 SNPs were conducted in the 23andMe, Genomics plc UK Biobank, FINRISK (PheWAS results release November 2016), and CHOP cohorts separately. Each SNP-phenotype association was tested independently (assuming an additive genetic model), using logistic regressions adjusted for age, gender, and principal components to adjust for population stratification. Genotyping batch and survey cohort were also included as covariates in the FINRISK PheWAS. We then performed two distinct analyses to (1) replicate known GWAS associations, and (2) to detect novel associations.

First, we meta-analyzed PheWAS results from the four cohorts, to investigate the ability of these cohorts to replicate known GWAS associations. After harmonizing the effect alleles across the cohorts, fixed effect meta-analyses were performed using PLINK[59]. $I^2$ statistic and manual review of the meta-analyzed results were used to evaluate heterogeneity between cohorts.

We then compared the meta-analysis association results with known significant SNP-phenotype associations from published GWAS, taking into account the statistical power to detect an association in the meta-analysis of the PheWAS results in the disease-agnostic cohorts.

Second, we meta-analyzed results from the four disease-agnostic cohorts together with available GWAS results in order to detect novel associations. Meta-analysis was performed using PLINK as described above. Meta-analysis results at the 145 shared phenotypes were then combined with cohort-specific phenotype results from the 25 SNPs, resulting in 27,762 tests in total. It is clear given the structure of this PheWAS and meta-PheWAS that the 27,762 tests are not independent tests, which requires thought about the most appropriate method to control for multiple testing correction. We have chosen two methods, one that provides an extremely, over-conservative multi-testing correction assuming independence (Bonferroni correction) and one less conservative method that has been shown to be robust to test dependency (Benjamini & Hochberg's False Discovery Rate (FDR)[64]. Benjamini and Yekutieli (2001) illustrated that the FDR procedure is robust to positive correlation amongst tests[65], therefore we have

chosen to use the standard Benjamini & Hochberg FDR procedure implemented in the p.adjust method in R. For defining significance in this study, we set a FDR threshold of 0.1, which corresponded to $P < 7 \times 10^{-4}$. The over-conservative significance threshold based on Bonferroni correction was $P = 0.05/27,762 = 1.8 \times 10^{-6}$. We note that Bonferroni correction ignores the correlation structure between the tested phenotypes or the fact that all the SNPs tested in this study are known to be associated with one or several phenotypes in published GWAS.

**Meta-analysis with UK Biobank v2 association results**. To further test the robustness of the putative novel associations identified in our study, we performed a meta-analysis of the 23andMe, FINRISK, CHOP, and published GWAS results for 41 SNP-phenotype pairs with association results released by Neale et al. from an analysis of the expanded UK Biobank cohort, consisting of up to 337,199 unrelated participants of European ancestry (referred to as UK biobank v2). In order to meta-analyze these UK Biobank v2 results, which had been obtained using linear regression models, with the PheWAS cohorts and GWAS results of the current study, which were obtained using logistic regression models, we performed a weighted $Z$ score meta-analysis. For each SNP-phenotype pair in each study $i$, we defined weights using the following equation:

$$W_i = 1 / \sqrt{(1/Na_i + 1/Nu_i)} \qquad (1)$$

where $Na_i$ and $Nu_i$ are the numbers of cases and controls in study $i$, respectively. For each SNP-phenotype pair, we then calculated the meta-analysis $Z$ score as follows:

$$Z = \sum (W_i * Z_i) / \sqrt{\sum (W_i^2)} \qquad (2)$$

$Z_i$ is the $Z$ score in study $i$, derived from the logistic or linear regression model. The UK Biobank GWAS results used in this analysis have been released by the Neale's lab under the following URL: https://sites.google.com/broadinstitute.org/ukbbgwasresults/home?authuser=0.

**Statistical power estimations**. We estimated statistical power to detect an association with known associated phenotypes using a formula adapted from Yang et al.[66], based on the published effect size in the most recently published GWAS, the frequency of the associated SNP risk allele in the 1000Genomes EUR population, the number of cases and controls in the disease-agnostic cohorts, and the following phenotype prevalence reported by the Centers for Disease Control and Prevention (https://www.cdc.gov): coronary artery disease, 5.8%; Crohn's disease, 0.2%; inflammatory bowel disease, 0.44%; myocardial infarction, 3%; multiple sclerosis, 0.09%; primary biliary cirrhosis, 0.04%; Parkinson's disease, 0.07%; psoriasis, 3%; rheumatoid arthritis, 0.6%; systemic lupus erythematosus, 0.2%; systemic scleroderma, 0.02%; type 1 diabetes, 0.5%; type 2 diabetes, 9%; ulcerative colitis, 0.24%; venous thromboembolism, 0.4%; vitiligo, 1%.

**Co-localization analyses**. To distinguish true pleiotropic effects from multiple associations at the loci that are explained by different causal SNPs (and potentially incriminating different causal genes), we used association summary statistics available from published GWAS and applied a Bayesian test implemented in the R package 'coloc' to assess co-localization, i.e., the probability of sharing causal genetic variants between pairs of apparent pleiotropic phenotypes using association summary statistics at the loci of interest[28]. Co-localization analysis at the *LGALS3* locus was performed using meta-analyzed PD GWAS summary statistics from 23andMe published elsewhere ($N$ cases = 4127, $N$ controls = 62,037)[26], and galectin-3 plasma pQTL results in 3562 blood donors[29]. Co-localization analysis at the *IFIH1* locus was performed using meta-analyzed SLE GWAS results from two independent published studies[67,68], meta-analyzed asthma GWAS summary statistics from 23andMe[69] and the Genomics plc UK Biobank (unpublished), and published UC GWAS summary statistics[70].

## Data availability

Full results from meta-analysis of the 23andMe, Gplc/UK Biobank, FINRISK and CHOP cohorts with published GWAS results are provided in the Supplementary Data 1. All summary statistics results from PheWAS in the individual cohorts can be requested to the respective authors.

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

## Acknowledgements

We thank the research participants from the 23andMe, UK Biobank, FINRISK, and CHOP cohorts for their contributions to this study. We would like to thank Jyoti Shah, Jennifer Pai, Mark Sharp, Hongjie Sun, and Ian Wallace for their input on phenotype mapping. We also thank Benjamin Neale's lab for generating and sharing the GWAS summary statistics from the expanded UK Biobank cohort, and Benjamin Neale for his input on meta-analyzing these results with standard disease association results derived from logistic regression. M.M. and H.H. are supported by a sponsored Research Grant from Merck, the Institutional Development Fund from CHOP and the eMERGE consortium grant 1U01HG006830-01 from the NHGRI.

## Author contributions

D.D., H.R., A.G.D.W., D.F.R., J.M., J.C.M., A.K.C., A.B., K.D.H.N., K.E. participated in the design and/or interpretation of the reported experiments or results. C.T., the 23andMe Research team and D.H. participated in the acquisition and/or analysis of the 23andMe data. C.S.F., C.C.A.S., C.V., M.E.W. and P.D. participated in the acquisition and/or analysis of the Genomics plc UK Biobank data and the Genomics plc GWAS data. M.A., H.M., E.K., M.R., J.H., N.B., S.J., D.G.M., V.S., S.R., M.J.D., and A.P. participated in the acquisition and/or analysis of the FINRISK data. M.M., P.M.A.S. and H.H. participated in the acquisition and/or analysis of the CHOP data. D.D. performed the phenotype mapping, meta-analysis and follow-up analyses. H.R. supervised the study, R.M.P., A.G.D.W., and C.S.F provided supervisory support. D.D. and H.R. wrote the manuscript.

## Additional information

**Competing interests:** D.D. and C.S.F. are employees at Merck Sharp and Dohme. D.F.R., J.M., J.C.M., A.K.C., A.B., A.G.D.W., R.M.P., and H.R. were employees at Merck Sharp and Dohme during the study and are now employed at other companies. C.T. and D.A. H. are employees of 23andMe Inc. C.S.F., C.C.A.S., M.E.W., and P.D. are employees of Genomics plc. C.V. was an employee of Genomics plc during this study. M.R. and J.H. are employees of Eisai. N.B. was an employee of Pfizer during this study. S.J. is an employee at Biogen. K.D.H.N. and K.E. were employees at Biogen during this study and are now employed at other companies. The remaining authors declare no competing interests.

## the 23andMe Research Team

Michelle Agee[2], Babak Alipanahi[2], Adam Auton[2], Robert K. Bell[2], Katarzyna Bryc[2], Sarah L. Elson[2], Pierre Fontanillas[2], Nicholas A. Furlotte[2], Bethann S. Hromatka[2], Karen E. Huber[2], Aaron Kleinman[2], Nadia K. Litterman[2], Matthew H. McIntyre[2], Joanna L. Mountain[2], Carrie A.M. Northover[2], Steven J. Pitts[2], J. Fah Sathirapongsasuti[2], Olga V. Sazonova[2], Janie F. Shelton[2], Suyash Shringarpure[2], Joyce Y. Tung[2], Vladimir Vacic[2] & Catherine H. Wilson[2]

