## [Peer Review File · Nature Communications]

Reviewer #1 (Remarks to the Author):

This is a nicely written manuscript that highlights the utility of PheWAS for target validation. The authors picked 25 variants in 19 genes that were causally associated with disease(s) and for which these genes are considered drug targets based on these associations. They then evaluated associations of these 25 SNPs across 4 cohorts for 145 mapped phenotypes and 1892 binary endpoints. They first replicate the associations of these 25 snps with the diseases that they were originally found to be associated with, replicating 70% of known associations. Then across all others – finding 10 new significant associations. The authors then provide assessment of 2 genes: PNPLA3 and IFIH1. The results are encouraging if not overly exciting. The authors do an excellent job of describing the potential and challenges of the approach.

Reviewer #2 (Remarks to the Author):

The authors conduct PheWAS meta-analyses on 25 selected SNPs across phenotypes obtained from four cohorts. The authors validate many known GWAS associations in these SNPs through their own GWAS and PheWAS analyses, and discover novel associations which may directly inform future work in drug development. The results are impressive: several of these associations are highlighted which corroborate or provide potential insight into known interactions between SNPs, proteins and clinical observations, which is an exciting step towards the practical application of GWAS and PheWAS results, although as the authors note, without further work the hypothesized function of these SNPs remains speculative. Overall, the analysis appears to follow conventional methods used in GWAS and PheWAS, although I would like to see the authors further address the combination of cohorts into the meta-analysis, as well as the attribution of pleiotropic effects to SNPs which affect similar phenotypes.

Major issues to be addressed

1. There appears to be a great potential for systematic biases in both genomes and phenomes between the study cohorts. The method for determination of European ancestry is different in each cohort: 23andMe uses an SVM and HMM approach; UK Biobank uses a combination of self-reporting and PCA; the FINRISK cohort is analyzed with MDS; and CHOP uses PCA. It may be worthwhile to demonstrate that these preprocessing approaches do not result in markedly different genetic cohorts, perhaps by conducting a single PCA across all cohorts, and visualizing clusters in the first two components. Additionally, if the CHOP cohort comprises data from pediatric patients, I question

the applicability of these data to traditionally age-related phenotypes listed among the novel associations, such as CAD, gallstones, high blood pressure, Parkinson's disease, and ever pregnant. Lastly, it would be useful to include more information about the manual mapping of common phenotypes between studies, including some example definitions within each cohort for phenotypes which were accepted or rejected as sufficiently similar. Some of the listed phenotypes such as HDL can be quantified on a continuous scale: did these phenotypes arrive pre-binarized in each cohort, or was some sort of thresholding applied across cohorts by the authors?

2. A more detailed description of the meta-analysis procedure is required. The authors state that "fixed effect and random effect meta-analyses were performed using PLINK," but no other explanation is offered. Are the results presented throughout the paper a result of the fixed effect or random effect analysis? Presumably a random effect treatment is required given the high potential for study heterogeneity mentioned above. However, the sizes of the datasets used in the meta-analysis vary greatly, from 600k to 12k, with further discrepancies in the percentage of respondents for each phenotype. This seems to result in the 23andMe dataset driving most of the discovered associations: most of the novel associations in Supplementary Table 4 are supported only by the 23andMe data. Therefore it seems a replication study for novel GWAS and PheWAS would help to verify the discovered associations. If the meta-analyses consistently showed directions of effect across most cohorts then it could be argued that this lessens the need for an independent replication, but as the results show, most of the associations are driven by a single cohort, and so it cannot necessarily be ruled out that associations are subject to study biases inherent in that cohort.

3. Among the challenges of conducting PheWAS mentioned briefly by the authors is the potential inability to distinguish true pleiotropy from non-genetic correlations or co-morbidities between phenotypes. However, the authors do not pursue this scenario beyond a call for more thorough investigation on a case-by-case basis. I believe further analysis on the existing data may elucidate the contribution of phenotypic correlations to the pleiotropic effects discovered. For example, methods such as LD score regression are capable of identifying the component of phenotypic correlations that can be attributed to shared genetic effects. Or the observed correlations between phenotypes within cohorts can simply be reported alongside their discovered associations with SNPs, so it is not assumed that SNP effects in multiple phenotypes are automatically attributed to pleiotropy.

4. The authors mention actions taken to minimize sample overlap between GWAS studies and between cohorts. However, I would like to see them address the possibility of including varying numbers of phenotype endpoints from single individuals within cohorts. This is particularly an issue with 23andMe data because certain individuals may be highly enthusiastic about filling out the online surveys, ensuring that their genomes are overrepresented among the phenotypes collected in the resulting cohort. Potentially, this may also bias the subsequent analyses if some of these individuals were also cases for multiple phenotypes, though this is admittedly unlikely. Additionally, if surveys for certain phenotypes have differing response rates, this may artificially increase the

correlation between phenotypes whose surveys are popular to fill out. It would be helpful to include a figure or table displaying a distribution over the number of surveys completed by distinct individuals, or the degree of sharing of individuals across phenotypes.

Minor issues to be addressed

5. There appear to be some discrepancies in the experiment parameters between the text and Figure 1. Page 8 states that across the four cohorts a total of 1892 binary phenotypes were extracted. However, this number appears to be 1683 in Figure 1, with 27611 tests. Page 9 further states that there are 1538 phenotypes and 27763 tests. It is possible that these numbers refer to different experimental conditions, but this is not immediately apparent from the text: if this is the case then they should be incorporated into the flow chart in Figure 1B.

6. The use of the term “real-world data” in the title and throughout the text is somewhat unclear, as this refers to genotypic data and phenotypic data consisting of a mixture of hospital records, hospital-administered questionnaires, and self-reporting through a web interface, none of which seems particularly unique compared to conventional GWAS. Perhaps the term refers to the large number of phenotypic variables in the data set, or to the more direct real-world applicability of the results, although I would argue that this is more of a description of the PheWAS method than the data itself. In any case, the term should be defined early in the text.

Reviewer #3 (Remarks to the Author):

The authors describe an investigation into the potential value of characterizing genetic effects across a broad phenotypic spectrum to inform drug discovery and development decisions. To do this, they identified 19 genes with one or more strong genetic associations and functional support for their influence on immune-mediated, cardiometabolic, or neurodegenerative diseases, and explored the impact of those variants on 1683 traits in up to 697,815 participants in four population samples with various forms of real world data. They demonstrated associations for most variants with the same traits there were selected for, and as anticipated, many others. They then explored how one might use those results in a drug discovery and development setting to support decisions around potential indications or safety risks.

By identifying potential drug targets having fairly convincing genetic associations to a potential clinical indication, they have genetic variants that capture some form of functional effect that can then be used to explore pleiotropic effects that could inform alternative indications or safety risks. In practice this would be a best case scenario in a drug discovery setting, but useful to illustrate the point of the study. They identify examples that demonstrate the potential value as well as potential pitfalls of such data, and approaches that could be taken to guard against them.

The authors explain their methods clearly, illustrate the results with appropriate figures and tables, and draw conclusions that are well supported by the results.

I don't have any major concerns. Below are a list of minor findings to address.

- * They refer to the 19 genes as candidate drug targets. In fact, at least 9 of them are reported drug targets and 4 in clinical development. It would be helpful to identify those in clinical development and the indications they are being developed for.

- * Note that capitalization of real-world data and phenome-wide associations studies is not used consistently (lines 73 and 78)

- * They mention PheWAS can be conducted with specific genetic variants (line 79). They can also be conducted with combinations of variants, as in TWAS-types of applications.

- * Line 98 refers to target validation at early stages of drug development. For consistency, drug development usually refers to clinical development. Are the authors comfortable that preclinical safety assessment falls under target validation?

- * Note "Trait" is used in Table S1, whereas "Phenotype" is used elsewhere.

- * Line 159 refers to proposed clinical indication. It wasn't clear to me what this referred to. I assume the trait for which the original genetic association was identified (Table S1). I suggest clarifying this.

Response to Reviewers' comments:

Reviewer #1 (Remarks to the Author):

This is a nicely written manuscript that highlights the utility of PheWAS for target validation. The authors picked 25 variants in 19 genes that were causally associated with disease(s) and for which these genes are considered drug targets based on these associations. They then evaluated associations of these 25 SNPs across 4 cohorts for 145 mapped phenotypes and 1892 binary endpoints. They first replicate the associations of these 25 snps with the diseases that they were originally found to be associated with, replicating 70% of known associations. Then across all others – finding 10 new significant associations. The authors then provide assessment of 2 genes: PNPLA3 and IFIH1. The results are encouraging if not overly exciting. The authors do an excellent job of describing the potential and challenges of the approach.

Response to Reviewer #1:

We thank Reviewer #1 for this very positive review.

Reviewer #2 (Remarks to the Author):

The authors conduct PheWAS meta-analyses on 25 selected SNPs across phenotypes obtained from four cohorts. The authors validate many known GWAS associations in these SNPs through their own GWAS and PheWAS analyses, and discover novel associations which may directly inform future work in drug development. The results are impressive: several of these associations are highlighted which corroborate or provide potential insight into known interactions between SNPs, proteins and clinical observations, which is an exciting step towards the practical application of GWAS and PheWAS results, although as the authors note, without further work the hypothesized function of these SNPs remains speculative. Overall, the analysis appears to follow conventional methods used in GWAS and PheWAS, although I would like to see the authors further address the combination of cohorts into the meta-analysis, as well as the attribution of pleiotropic effects to SNPs which affect similar phenotypes.

Response to Reviewer #2:

We thank Reviewer #2 for the very helpful comments.

Major issues to be addressed

1. There appears to be a great potential for systematic biases in both genomes and phenomes between the study cohorts. The method for determination of European ancestry is different in each cohort: 23andMe uses an SVM and HMM approach; UK Biobank uses a combination of self-reporting and PCA; the FINRISK cohort is analyzed with MDS; and CHOP uses PCA. It may be worthwhile to demonstrate that these preprocessing approaches do not result in markedly different genetic cohorts, perhaps by conducting a single PCA across all cohorts, and visualizing clusters in the first two components.

We agree that meta-analyzing results across differently ascertained cohorts may introduce systematic biases and had highlighted this as a potential limitation of our approach in the

discussion. Since our analyses rest on the summary statistics available to us, a single combined PCA analysis across all cohorts is unfortunately not feasible.

Nevertheless, for several reasons we are confident that the ancestry bias in our data is minimal: First, the methods used by the respective cohorts to assess population stratification are all standard in the field. Second, GWAS consortia frequently pool data from multiple sources in a very similar manner, with thus far little evidence for significant artifacts relative to previous studies [for examples see recent GWAS meta-analyses for Parkinson's disease (PMID: 28892059) or migraine (PMID: 27322543) which combined 23andMe data with data from other GWAS cohorts]. Third, at $P < 0.05$, significance cutoff often used in replication analysis, we replicate 75% of previously reported associations ("positive controls") that are adequately powered ($\beta \geq 0.8$), with effect sizes across the four cohorts consistent with previously published data.

In order to better visualize the robustness of our data, we have added two new Supplementary Figures (**Supplementary Figure 2** and **Supplementary Figure 3**) that illustrate the correlation of published GWAS effect sizes with the effect sizes observed in our meta-PheWAS (Supplementary Figure 2), as well as in each of the four individual cohorts (Supplementary Figure 3). As shown in Supplementary Figure 2, all but two of the 39 known GWAS associations replicating at $P < 0.05$ in our meta-PheWAS show consistent directions of effects with the published effect sizes. In addition, Supplementary Figure 2 shows the near linear relationship between the point estimates of the effect sizes in the published GWAS and those in the meta-PheWAS, with a regression coefficient $r^2 = 0.68$. Supplementary Figure 3 shows that the distribution is similar for each of the four individual cohorts, supporting that heterogeneity between the cohorts is limited.

Additionally, if the CHOP cohort comprises data from pediatric patients, I question the applicability of these data to traditionally age-related phenotypes listed among the novel associations, such as CAD, gallstones, high blood pressure, Parkinson's disease, and ever pregnant.

Our intention when including the pediatric CHOP cohort was twofold: First, to further expand the spectrum of phenotypes tested and identify putative associations to diseases with an earlier manifestation. Indeed, 9 putative associations reaching $FDR < 0.1$ are with phenotypes tested only in CHOP and will require replication in independent cohorts capturing these phenotypes (see **Supplementary Table 5**). Second, through the meta-PheWAS, we aimed to increase power for phenotypes that were captured in more than a single cohort. We found 15 novel associations reaching $FDR < 0.1$ in the meta-PheWAS of mapped phenotypes that include CHOP results (**Supplementary Table 5**), further demonstrating applicability of this cohort. Since the CHOP cohort is the smallest cohort in our study and typically does not capture later-onset diseases (due to the absence of cases) it is unlikely that this cohort introduces significant biases. Instead, the CHOP cohort added truly relevant information for several SNP-phenotype pairs. This is probably best illustrated at the case of *IFIH1* and asthma where the point estimate of the effect size in the CHOP cohort was consistent with the effect size observed in the other cohorts and contributed to the significant association in the meta-PheWAS (see **Figure 3A**). Furthermore, 30 of the 32 known associations tested in CHOP (used as positive controls) showed consistent directions of effects with the published effect sizes, with an overall linear relationship between ORs in published

GWAS vs CHOP PheWAS demonstrated by a regression coefficient $r^2 = 0.64$ (see **Supplementary Figure 3D**).

Lastly, it would be useful to include more information about the manual mapping of common phenotypes between studies, including some example definitions within each cohort for phenotypes which were accepted or rejected as sufficiently similar.

We have added a new Supplementary Table (**Supplementary Table 2**) to better illustrate the phenotype mapping process at the case of several selected examples. In addition, we now provide the specific terms used by 23andMe, UK biobank, FINRISK and CHOP, respectively, that were used to define mapped endpoints for our meta-analyses in the **Supplementary Datasheet**.

We note that in the course of this project we had tested tools to assist with phenotype mapping, including KUSP (<https://kusp.factbio.com/kusp>) and Tamr (<https://www.tamr.com>). However, at the time of our analyses, these tools did not perform convincingly, for instance in mapping acronyms (e.g. "AMD") or popular terms (e.g. "heart attack") to clinical terms. Since these and other tools are constantly improving, it can be expected that future studies like ours may benefit from semi-automated phenotype mapping that will further improve the manual mapping applied here.

Some of the listed phenotypes such as HDL can be quantified on a continuous scale: did these phenotypes arrive pre-binarized in each cohort, or was some sort of thresholding applied across cohorts by the authors?

As we state in the manuscript (Methods, Results, Table 2), all phenotypes tested in our study are binary traits. All traits, including "high LDL cholesterol", "low HDL" or "high blood pressure", are derived from questionnaires/self-reports or ICD codes and were not available as laboratory measurements.

We have modified the following sentence in the manuscript to clarify this point:

Line 389: "All endpoints were derived from questionnaires or ICD codes (including endpoints like "high cholesterol" or "high blood pressure"). No quantitative laboratory measurements were included the study."

2. A more detailed description of the meta-analysis procedure is required. The authors state that "fixed effect and random effect meta- analyses were performed using PLINK," but no other explanation is offered. Are the results presented throughout the paper a result of the fixed effect or random effect analysis? Presumably a random effect treatment is required given the high potential for study heterogeneity mentioned above. However, the sizes of the datasets used in the meta-analysis vary greatly, from 600k to 12k, with further discrepancies in the percentage of respondents for each phenotype. This seems to result in the 23andMe dataset driving most of the discovered associations: most of the novel associations in Supplementary Table 4 are supported only by the 23andMe data. Therefore it seems a replication study for novel GWAS and PheWAS would help to verify the discovered associations. If the meta-analyses consistently showed directions of effect across most cohorts then it could be argued that this lessens the

need for an independent replication, but as the results show, most of the associations are driven by a single cohort, and so it cannot necessarily be ruled out that associations are subject to study biases inherent in that cohort.

As highlighted in **Figure 1**, the phenotypes captured in each of the four cohorts varied considerably. Consequently, many endpoints were unique to only one cohort, limiting availability of endpoints for meta-analyses. Due to the unique nature of self-reported phenotypes as well as its massive sample size, the 23andMe cohort was the richest with respect to endpoints meeting our criteria of sufficient number of cases per phenotype to be included in our analysis (N cases >20). We have added the following statement to the Results section to address this point and emphasize the need for independent replication of novel associations derived from a single cohort:

Line 165: "Using a less stringent significance threshold of $FDR < 0.1$ ($P < 7 \times 10^{-4}$) previously applied in PheWAS²⁵, we identified 72 distinct putative novel associations (**Fig. 2, Supplementary Table 5 and Supplementary Datasheet**). Of these, 31 were with mapped phenotypes and were obtained from meta-analyzing results from at least two cohorts, and 41 were supported by a single cohort (and thus will require independent replication)".

The meta-analysis results reported in our study are from fixed-effect meta-analyses. All four cohorts were restricted to individuals of European ancestry and, as shown in **Supplementary Fig. 2 and Supplementary Fig. 3**, limited genetic heterogeneity between the cohorts is expected. Since our study includes four cohorts with large differences in sample sizes, a random effect meta-analysis would give more weight to small studies and would strongly affect power in the meta-analyses [e.g., see Jackson and Turner, 2017. Power analysis for random-effects meta-analysis. Res Synth Methods 8(3):290-302. PMID: 28378395]. On the other hand, a fixed-effect meta-analysis gives more weight to larger studies which provide more precise estimates of the effect sizes. To evaluate the heterogeneity in the fixed-effect meta-analysis results, we used the I^2 statistic. Out of the 31 putative novel associations that were obtained from meta-analysis of ≥ 2 cohorts, 24 (77%) showed low-to-moderate heterogeneity with an $I^2 < 40\%$. We manually reviewed the results for the 7 associations with $I^2 > 40\%$. Six showed a p-value in the meta-analysis that was more significant than the p-values in the individual cohorts. The only exception is the association of *GDF15* SNP rs763361 with high blood pressure that was significant in 23andMe ($P_{23andMe} = 6.4 \times 10^{-10}$), but appeared less significant when meta-analyzed with UK Biobank results ($P_{META} = 7.6 \times 10^{-9}$). We have added the I^2 values to **Supplementary Table 5** and the **Supplementary Datasheet**. We have also added a new Supplementary Figure (**Supplementary Fig. 5**) that shows the effect sizes in the individual cohorts for each of the novel associations obtained for meta-analysis of ≥ 2 cohorts. We have made the following modifications to the text to highlight these changes:

Line 176: "The 31 novel associations with mapped phenotypes showed limited evidence of heterogeneity between the PheWAS cohorts (**Supplementary Fig. 5**). Twenty-four (77%) showed an $I^2 < 40\%$. Manual review of the results showed that only one of the seven associations with $I^2 > 40\%$, the *GDF15* rs17724992 association with high blood pressure, was less significant in the meta-analysis than in the individual cohorts and was more likely to be a false-positive association ($P_{23andMe} = 6.4 \times 10^{-10}$, $OR_{23andMe} = 0.96$; $P_{Gplc/UK\ Biobank} = 0.58$, $OR_{Gplc/UK\ Biobank} = 0.99$; $P_{META} = 7.6 \times 10^{-9}$, $OR_{META} = 0.97$) (**Supplementary Fig. 5B**)".

Line 488: "After harmonizing the effect alleles across the cohorts, fixed effect meta-analyses were performed using PLINK⁵⁷. I^2 statistic and manual review of the meta-analyzed results were used to evaluate heterogeneity between cohorts."

3. Among the challenges of conducting PheWAS mentioned briefly by the authors is the potential inability to distinguish true pleiotropy from non-genetic correlations or co-morbidities between phenotypes. However, the authors do not pursue this scenario beyond a call for more thorough investigation on a case-by-case basis. I believe further analysis on the existing data may elucidate the contribution of phenotypic correlations to the pleiotropic effects discovered. For example, methods such as LD score regression are capable of identifying the component of phenotypic correlations that can be attributed to shared genetic effects. Or the observed correlations between phenotypes within cohorts can simply be reported alongside their discovered associations with SNPs, so it is not assumed that SNP effects in multiple phenotypes are automatically attributed to pleiotropy.

As we discuss extensively in the Results and Discussion sections, distinguishing apparent pleiotropy (due to comorbidities, confounders (e.g. medication use) or lack of co-localization) from true pleiotropy is a key challenge in interpreting PheWAS results that has largely been overlooked thus far. Although the lack of access to individual-level data limited our ability to investigate this question more systematically, we present results adjusted for elevated liver test (**Supplementary Table 6**) and autoimmune diseases (**Supplementary Table 7**) and perform co-localization analyses to strengthen evidence for pleiotropic effects at the *PNPLA3* and *IFIH1* loci and against pleiotropy at the *LGALS3* locus (**Fig. 3** and **Supplementary Fig. 4**).

As suggested by the reviewer, we have added a new Supplementary Figure (**Supplementary Fig. 7**) to illustrate the phenotype correlations in the 23andMe data between all known and novel associations. This should enable readers to estimate the potential contribution of phenotype co-associations/co-morbidities to the PheWAS results. These results should be considered with caution as there are many reasons why a pair of phenotypes might be correlated. We also note that confirming or excluding true pleiotropy is complex and requires thorough investigation of each signal.

4. The authors mention actions taken to minimize sample overlap between GWAS studies and between cohorts. However, I would like to see them address the possibility of including varying numbers of phenotype endpoints from single individuals within cohorts. This is particularly an issue with 23andMe data because certain individuals may be highly enthusiastic about filling out the online surveys, ensuring that their genomes are overrepresented among the phenotypes collected in the resulting cohort. Potentially, this may also bias the subsequent analyses if some of these individuals were also cases for multiple phenotypes, though this is admittedly unlikely. Additionally, if surveys for certain phenotypes have differing response rates, this may artificially increase the correlation between phenotypes whose surveys are popular to fill out. It would be helpful to include a figure or table displaying a distribution over the number of surveys completed by distinct individuals, or the degree of sharing of individuals across phenotypes.

We agree with the reviewer that assessing the consequences of variation in numbers of phenotypes per individual on PheWAS outcomes would be a very interesting experiment. This should probably be best addressed in an independent manuscript more dedicated towards PheWAS methodology. Previous studies from the 23andMe research team provide confidence that variation in phenotype availability does not massively distort detection of true associations (see e.g., Tung et al., 2011.

Efficient replication of over 180 genetic associations with self-reported medical data. PLOS One 6:e23473; PMID: 21858135). Phenotype correlation scores in the 23andMe data included in our study (see new **Fig. S7**) largely support the expected relationship between phenotypes.

Minor issues to be addressed

5. There appear to be some discrepancies in the experiment parameters between the text and Figure 1. Page 8 states that across the four cohorts a total of 1892 binary phenotypes were extracted. However, this number appears to be 1683 in Figure 1, with 27611 tests. Page 9 further states that there are 1538 phenotypes and 27763 tests. It is possible that these numbers refer to different experimental conditions, but this is not immediately apparent from the text: if this is the case then they should be incorporated into the flow chart in Figure 1B.

N=1,892 refers to the total number of endpoints across the four cohorts before phenotype mapping (Table 2). N=1,683 refers to the number of endpoints tested in our analysis after phenotype mapping (145 mapped endpoints for meta-analysis + 1538 cohort-specific unmapped endpoints). To clarify this point and improve readability of the manuscript, we have modified the Results section as follows:

- We have removed the following line (previously line 134): "Together the four RWD cohorts allowed association testing for a total of 1,892 binary endpoints"

- We have modified line 127 as follows: "Manual phenotype mapping identified 145 distinct clinical endpoints that were tested in two or more cohorts, enabling meta-analyses in up to 697,815 individuals (Fig. 1B, Supplementary Table 2 and Supplementary Table 3). As illustrated in Fig. 1C, these 145 mapped phenotypes represent a broad spectrum of disease categories and, as typically observed in disease-agnostic cohorts, show significant variability in the case:control ratios, both within and between cohorts. In addition, PheWAS in the four cohorts provided associations results for 1,538 cohort-specific unmapped endpoints, leading to a total of 1,683 endpoints included in our analysis."

N=27,611 refers to the number of tests in the meta-PheWAS including only the 4 disease-agnostic cohorts. N=27,763 refers to the number of tests when combining the meta-PheWAS results with GWAS results from 44 studies. We have modified **Fig. 1B** to clarify this point.

6. The use of the term "real-world data" in the title and throughout the text is somewhat unclear, as this refers to genotypic data and phenotypic data consisting of a mixture of hospital records, hospital-administered questionnaires, and self-reporting through a web interface, none of which seems particularly unique compared to conventional GWAS. Perhaps the term refers to the large number of phenotypic variables in the data set, or to the more direct real-world applicability of the results, although I would argue that this is more of a description of the PheWAS method than the data itself. In any case, the term should be defined early in the text.

The term "real-world data cohort" has been coined to distinguish cohorts that are primarily established for the purpose of research from cohorts for which data has been generated primarily for other reasons (e.g., during routine clinical care). To avoid confusion, we have replaced the term "real-world data cohort" by "disease-agnostic cohort with extensive health information" throughout the text.

Reviewer #3 (Remarks to the Author):

The authors describe an investigation into the potential value of characterizing genetic effects across a broad phenotypic spectrum to inform drug discovery and development decisions. To do this, they identified 19 genes with one or more strong genetic associations and functional support for their influence on immune-mediated, cardiometabolic, or neurodegenerative diseases, and explored the impact of those variants on 1683 traits in up to 697,815 participants in four population samples with various forms of real world data. They demonstrated associations for most variants with the same traits there were selected for, and as anticipated, many others. They then explored how one might use those results in a drug discovery and development setting to support decisions around potential indications or safety risks.

By identifying potential drug targets having fairly convincing genetic associations to a potential clinical indication, they have genetic variants that capture some form of functional effect that can then be used to explore pleiotropic effects that could inform alternative indications or safety risks. In practice this would be a best case scenario in a drug discovery setting, but useful to illustrate the point of the study. They identify examples that demonstrate the potential value as well as potential pitfalls of such data, and approaches that could be taken to guard against them.

The authors explain their methods clearly, illustrate the results with appropriate figures and tables, and draw conclusions that are well supported by the results.

I don't have any major concerns. Below are a list of minor findings to address.

Response to Reviewer #3:

We thank reviewer #3 for these very positive comments.

* They refer to the 19 genes as candidate drug targets. In fact, at least 9 of them are reported drug targets and 4 in clinical development. It would be helpful to identify those in clinical development and the indications they are being developed for.

We fully agree and thank the reviewer for this suggestion. A new **Table 1** in the main manuscript now summarizes the genetic support and current drug development status (based on Citeline's Pharmaprojects database) for each of the 19 genes investigated in our study.

* Note that capitalization of real-world data and phenome-wide associations studies is not used consistently (lines 73 and 78)

We have modified "Phenome-wide Associations Studies" to "phenome-wide association studies" in line 71 for consistency. We have further replaced the term real-world data (RWD) by "disease-agnostic cohort with extensive health information" throughout the text to avoid misconception.

* They mention PheWAS can be conducted with specific genetic variants (line 79). They can

also be conducted with combinations of variants, as in TWAS-types of applications.

We have modified the respective sentence as follows:

Line 71: *“PheWAS are an unbiased approach to test for associations between a specific genetic variant, or more recently, combinations of variants, and a wide range of phenotypes in large numbers of individuals^{7, 15, 16}”.*

* Line 98 refers to target validation at early stages of drug development. For consistency, drug development usually refers to clinical development. Are the authors comfortable that preclinical safety assessment falls under target validation?

In our study we applied PheWAS to assess efficacy and safety of drug targets at both, preclinical and clinical stages (see new **Table 1**). To avoid confusion, we have now modified this sentence as follows:

Line 90: *“Here, we have tested the hypothesis that PheWAS can inform target validation at early stages of drug discovery.”*

* Note "Trait" is used in Table S1, whereas "Phenotype" is used elsewhere.

We have replaced the term "trait" by "phenotype" in the main text and Supplementary Table 1 for consistency.

* Line 159 refers to proposed clinical indication. It wasn't clear to me what this referred to. I assume the trait for which the original genetic association was identified (Table S1). I suggest clarifying this.

We have added **Table 1** to clarify this point. We have also modified this sentence as follows:

Line 156: *“We next investigated whether meta-PheWAS across the four cohorts could identify novel associations to support the proposed clinical indication(s) (derived from established genetic associations, see **Table 1**), suggest alternative indications for drug repositioning, or uncover potential target-related ADEs.”*

Further revisions:

We have added two co-authors.

Reviewer 2
comment 1:

1. Merging artifacts:

- a. The authors reject the concern of population artifacts based on
 - i. Standards in GWAS meta analysis (labeled “first” & “second”)
This argument does not hold. GWAS meta analyses relies on genomewide sanity checks, first and foremost being the analysis of the distribution of the observed statistical signals (qq-plot). This battery of tools is not available when conducting PheWAS that is not genomewide. It is incumbent on the authors to come up with an alternative.
 - ii. Success in replication (labeled “third”).
This argument is not supported by the data as provided by the authors. Supp. Table 4 contains multiple entries that are fail replication despite listed power of 1 (rs13391356 for IBD at experiment-wide significance, rs11145766 for IBD at nominal or FDR<0.1 rs4077515 for IBD at FDR<0.1, same SNP for the same phenotype (listed again???) for experiment-wide significance etc.) . This significantly rejects the author’s claim. I assume the fault is much due to over optimistic power calculations, but that’s for the authors to figure out. Supp. Sig. 3 only emphasizes the SNPs (most visibly in panel A) whose cohort-specific effect sizes are statistically inconsistent.
- b. Inclusion of the CHOP cohort in the Meta PheWAS:
The authors make three rebuttals:
 - i. Phenotypes from only CHOP:
This means that for these phenotypes are not in a meta PheWAS, so the argument does not address the issue.
 - ii. CHOP in increases some novel association signals, e.g. IFIH1/asthma
Per Fig. 3A, CHOP is consistent with the null of OR=1 for this association, so the example is unsupported. Given the differences between childhood asthma and adult asthma, this is a particularly unconvincing example.
 - iii. Known associations tend to have the same sign/replicate
I accept this argument, even if it is not quantitatively made. A full argument would involve consistency of the CHOP effect sizes with the published ones.
- c. Conversion of phenotype terms – Supp. Table 2 is appreciated
- d. Quantitative thresholds – rebuttal accepted.

2. Lack of replication for single-cohort results:

Thanks for the admission that most (41/72) of the claimed results do not meet the scientific standard in the field: a result with replication. Even the 31 hits at FDR<0.1 do not meet the standards of study-wide significance + replication, established for GWAS as a lesson from the falsities-plagued era of

candidate gene studies. The work should report in the abstract only results that meet scientific standards.

3. Rebuttal accepted, though it highlights the number of independent hits (and tests) to be smaller than claimed, thus the FDR threshold is invalid and should be recomputed.
4. Rebuttal accepted
5. Minor comments – rebuttal accepted

Reviewer #3 (Remarks to the Author):

I have reviewed the response and changes made to the manuscript and supplementary materials. The authors have addressed each of my concerns and improved the manuscript overall. The addition of Supplementary Figures 2 and 3 are particularly insightful.

Reviewer #2 (Remarks to the Author):

1. Merging artifacts:

a. The authors reject the concern of population artifacts based on

i. Standards in GWAS meta analysis (labeled “first” & “second”) This argument does not hold. GWAS meta analyses relies on genomewide sanity checks, first and foremost being the analysis of the distribution of the observed statistical signals (qq-plot). This battery of tools is not available when conducting PheWAS that is not genomewide. It is incumbent on the authors to come up with an alternative.

As we highlight in our previous response to the reviewer, the methods we use to adjust for population stratification are standard in meta-analyses from GWAS summary statistics. Studies that detail QC steps and association results for 23andMe, UK biobank, FINRISK and CHOP cohorts have been published previously (Pickrell et al. 2016, Nat Genet, PMID: 27182965; Bycroft et al. 2017, doi: <http://dx.doi.org/10.1101/166298>; Zhao et al. 2017, PMID: 28869590; Gormley et al. 2018, PMID: 29731251; Grant et al. 2014, PMID: 19656524). This includes the study from Pickrell et al. published in Nature Genetics (PMID: 27182965) that describes GWAS results for 17 phenotypes derived from 23andMe, including asthma and hypothyroidism, two of the traits with significant associations in our study. Several publications have jointly analyzed data from these cohorts with other cohorts for genetic association testing without indications for systematic biases (e.g. Nalls et al. 2014, PMID: 25064009; Zhao et al, 2017, PMID: 28869590; Jostins et al. 2012, PMID: 23128233). These data, together with the high concordance rate of our findings with previous GWAS signals in Europeans (**Supplemental Figures 1-3**) and between the cohorts (**Table 1**, **Supplementary Figure 5**, as well as new **Supplementary Table 6**) make systematic biases from marked genetic differences between the cohorts highly unlikely.

ii. Success in replication (labeled “third”).

This argument is not supported by the data as provided by the authors. Supp. Table 4 contains multiple entries that are fail replication despite listed power of 1 (rs13391356 for IBD at experiment-wide significance, rs11145766 for IBD at nominal or FDR<0.1 rs4077515 for IBD at FDR<0.1, same SNP for the same phenotype (listed again???) for experiment-wide significance etc.). This significantly rejects the author’s claim. I assume the fault is much due to over optimistic power calculations, but that’s for the authors to figure out. Supp. Sig. 3 only emphasizes the SNPs (most visibly in panel A) whose cohort-specific effect sizes are statistically inconsistent

As we describe in Results (**I.138-142**) and our previous response to the reviewer, 75% and 67% of established, sufficiently powered GWAS-associations tested in our study replicate at $P < 0.05$ and $FDR < 0.1$, respectively. These high replication rates surpass replication rates of previous PheWAS (e.g., 66% at $P < 0.05$ in Denny et al. 2013, PMID: 24270849) and support that our study is adequately corrected for population stratification. Consistency of effect size point estimates between published GWAS and our PheWAS results is generally high, irrespective of whether an association has been found for a respective SNP or not, as best exemplified by R^2 values between 0.58 and 0.94 across the four cohorts (**SupplementaryFigure 3**). In the current version of our manuscript we now also add replication results for the proposed novel associations that further support this point (see our comments to this Reviewer’s point 2. below).

We also highlight in Results (I.147-150) that 8 of the 11 associations (driven by 5 distinct SNPs) that failed to replicate despite in principle sufficient power are associations with inflammatory bowel disease (IBD) and its subtypes Crohn's disease (CD) and ulcerative colitis (UC). For 7 of the 8 non-replicating IBD/CD/UC loci, directionality in our meta-PheWAS matches that of previous GWAS, although effect sizes of the risk alleles are lower in our current study (see **Supplementary Table 3** and **Rebuttle Figure 1** below). Statistical power estimations were performed with routine analysis tools, and up-to-date disease prevalence information was obtained from the Centers of Disease Control (with prevalence rates for IBD, CD and UC of 0.44%, 0.2% and 0.24%, respectively; see Methods, I.557-567) and are thus unlikely to be particularly over-optimistic. A much more likely scenario is that the closely related and clinically difficult to discern inflammatory disorders IBD, CD and UC are sub-optimally ascertained in real-world settings, which we discuss in I.358-360 and which has been reported previously (e.g. Silverberg et al., 2001; PMID: 11709510).

We thank the reviewer for pointing out an oversight in **Supplementary Table 4**. An association between rs4077515 and IBD had erroneously been listed twice, while the lower line was supposed to refer to the next SNP on the list, rs763361 in *CD226*. This typo has now been corrected.

OR

Rebuttal Figure 1: Effect sizes of known associations with CD, UC and IBD that fail to replicate in our meta-PheWAS. Published ORs are shown as red circles. Odds ratios and 95% confidence intervals in the meta-PheWAS are shown as squares and lines.

- b. Inclusion of the CHOP cohort in the Meta PheWAS: The authors make three rebuttals:
- i. Phenotypes from only CHOP: This means that for these phenotypes are not in a meta PheWAS, so the argument does not address the issue.

This argument addressed what we assumed to be part of the reviewer's question, namely whether CHOP data can be used *in general* for traditional GWAS analyses. As discussed in our previous response, we found 9 CHOP-specific potential novel associations (FDR<0.1) with the following phenotypes that typically are either age-related or poorly captured in published GWAS: acide-base balance disorder, anorexia, hypoventilation, obstructive sleep apnea, pituitary hypofunction, psoriatic arthropathy, speech and language disorder, systolic/diastolic heart failure, tachycardia.

ii. CHOP increases some novel association signals, e.g. *IFIH1*/asthma. Per Fig. 3A, CHOP is consistent with the null of OR=1 for this association, so the example is unsupported. Given the differences between childhood asthma and adult asthma, this is a particularly unconvincing example.

As shown in **Figure 3A**, although the *IFIH1* SNP is not associated with asthma at $P < 0.05$ in the CHOP cohort alone, the point estimate of the effect size of rs1990760-C in CHOP is highly consistent with the other cohorts in this study. The CHOP cohort does contribute to the meta-analysis association signal, as reflected by a stronger association when CHOP is included ($P = 6.5 \times 10^{-8}$; OR=1.0377; $I^2 = 0$; N=6 studies) than when it is not ($P = 1.4 \times 10^{-7}$; OR=1.0376; $I^2 = 0$; N=5 studies). We also note that the *IFIH1*/asthma association is further strengthened when our results are meta-analyzed with the expanded UK Biobank “v2” dataset of N=337,199 participants (GpIC/UK Biobank results excluded), reaching $P_{\text{meta}} = 2.01 \times 10^{-8}$ (see also below). Interestingly, despite ascertainment of this phenotype from more than 337,000 participants, the number of asthma cases in the latest release of UK biobank alone has been insufficient to reveal the *IFIH1*/asthma association at genome-wide significance ($\beta = 0.35$) (De Boever et al., Nature Communications (2018)9:1612; Emdin et al., Nature Communications (2018)9:1613). We estimate that revealing this association would have necessitated 59,908 asthma cases, about 1.5-times the case number in the current, expanded UK biobank “v2” dataset ($n = 39,049$; see also our new **Supplementary Table 6** and below). This strongly supports our approach of meta-PheWAS across multiple cohorts to identify significant association signals.

iii. Known associations tend to have the same sign/replicate. I accept this argument, even if it is not quantitatively made. A full argument would involve consistency of the CHOP effect sizes with the published ones.

As a previous response to this reviewer, we added **Supplementary Figure 3D** to highlight the overall linear relationship between CHOP ORs and ORs from published GWAS. The best quantitative measurement to demonstrate consistency in the effect sizes across all SNPs tested is the observed positive regression coefficient ($R^2 = 0.64$). For putative novel associations, effect sizes in CHOP relative to the other cohorts and meta-analysis outcomes are visualized in **Supplementary Figure 5**.

c. Conversion of phenotype terms – Supp. Table 2 is appreciated

d. Quantitative thresholds – rebuttal accepted.

2. Lack of replication for single-cohort results:

Thanks for the admission that most (41/72) of the claimed results do not meet the scientific standard in the field: a result with replication. Even the 31 hits at $FDR < 0.1$ do not meet the standards of study-wide significance + replication, established for GWAS as a lesson from the falsities-plagued era of candidate gene studies. The work should report in the abstract only results that meet scientific standards.

We fully agree that replication is desirable and in our revised manuscript have now addressed this point through replication in the expanded UK Biobank cohort (see below).

However, we first want to point out that most PheWAS to date are single-cohort studies and replication is not yet standard in the field. Even very recent prominent PheWAS either did not attempt to replicate novel associations (e.g., Verma et al. 2018, AJHG; PMID: 29606303), or

have tried to circumvent the issue by randomizing their total sample size into a discovery and a replication cohort (e.g., Verma et al. 2016, BMC Med Genomics; PMID: 27535653), which does not account for potential cohort-specific biases. By exploring the potential for systematic PheWAS meta-analyses across multiple cohorts, our study attempts to address exactly this gap. While 31 of the novel associations in our "discovery" analysis meet or exceed generally accepted significance thresholds for PheWAS and are supported by more than one cohort (see e.g. **Supplementary Figure 4** and **Supplementary Table 5**), others are with cohort-specific endpoints. One frequent reason for this is that the associated endpoints are captured only in a single cohort (see **Figure 1** for visualization and **Supplementary Datasheet** for source data), and/or mapping endpoints across cohorts poses challenges that we discuss extensively. In response to this reviewer's previous comments, we were careful not to overstate our findings and carefully pointed out the limitations, e.g. in **I.161-168**:

"Overall, 27,763 association tests (across 145 harmonized and 1,538 cohort-specific endpoints) resulted in 9 putative novel associations reaching study-wide significance after Bonferroni correction ($P < 1.8 \times 10^{-6}$) (Table 3). Using a less stringent significance threshold of $FDR < 0.1$ ($P < 7 \times 10^{-4}$) previously applied in PheWAS, we identified 71 distinct putative novel associations (Fig. 2, Supplementary Table 5 and Supplementary Datasheet). Of these, 30 were with mapped phenotypes and were obtained from meta-analyzing results from at least two cohorts, and 41 were supported by a single cohort (and thus will require independent replication)".

To make this point even clearer, we now highlight in our current manuscript whether a signal is supported by only a single cohort, or whether it has been derived from meta-analysis in ≥ 2 cohorts and is thus independently replicated. For instance, **I.240-243** now reads:

"The meta-PheWAS also revealed significant associations between rs738409-G and decreased risk for high cholesterol ($OR_{meta} = 0.96$, $P_{meta} = 1.6 \times 10^{-7}$; $P_{meta_v2} = 1.1 \times 10^{-8}$) and the intake of cholesterol-lowering medications ($OR_{23andMe} = 0.97$, $P_{23andMe} = 2 \times 10^{-4}$; $P_{meta_v2} = 2.8 \times 10^{-5}$)".

In the current version of our manuscript, we further strengthen our findings through replication in the expanded UK Biobank cohort (referred to as "UK biobank v2"). 41/71 potential novel associations with $FDR < 0.1$, and 8/9 novel associations reaching Bonferroni correction were with phenotypes tested through GWAS in 337,119 participants of European ancestry in the UK Biobank as recently released by Neale *et al.* (<http://www.nealelab.is/blog/2017/7/19/rapid-gwas-of-thousands-of-phenotypes-for-337000-samples-in-the-uk-biobank>). We meta-analyzed these "UK biobank v2" results obtained using linear regression models with the three independent PheWAS cohorts 23andMe, FINRISK and CHOP (Genomics plc UK Biobank results were excluded from this analysis) and GWAS results obtained using logistic regression models, using a weighted Z-score meta-analysis approach. Out of the 41 putative novel associations tested in this replication analysis, 16 showed $P < 0.05$ in UK Biobank v2 with consistent direction of effect and leading to increased significance in meta-analysis with the PheWAS cohorts and the GWAS results (new **Supplementary Table 6**). An additional seven potential novel associations showed increased significance in this meta-analysis despite $P > 0.05$ in UK Biobank v2, largely due to small number of cases and lack of statistical power in UK Biobank v2. Overall, UK Biobank v2 strengthened all 8 novel associations reaching study-wide significance after Bonferroni correction and 23/41 (56%) of the potential novel associations observed in our study, including 8 associations that were based on results from only one PheWAS cohort in the "discovery" analysis.

Among others, replication strengthened associations that we had previously highlighted in the manuscript, such as the *PNPLA3* rs738409-G associations with acne ($P_{23andMe} = 1.5 \times 10^{-11}$; $P_{meta_v2} = 7.3 \times 10^{-12}$), high cholesterol ($P_{meta} = 1.6 \times 10^{-7}$; $P_{meta_v2} = 1.1 \times 10^{-8}$), cholesterol-lowering medication ($P_{23andMe} = 2 \times 10^{-4}$; $P_{meta_v2} = 2.8 \times 10^{-5}$), gout ($P_{meta} = 4.1 \times 10^{-5}$; $P_{meta_v2} = 3.9 \times 10^{-9}$), and

gallstones ($P_{\text{meta}}=2.7 \times 10^{-4}$; $P_{\text{meta}_v2}=1.5 \times 10^{-5}$), and the *IFIH1* rs1990760-C association with asthma ($P_{\text{meta}}=9.0 \times 10^{-8}$, $P_{\text{meta}_v2}=2 \times 10^{-8}$).

The new results from replication analyses have been added to our current manuscript as new **Supplementary Table 6**. In the revised manuscript, replication results are discussed in the following sections:

Results section , I.183-200:

"Replication of potential novel associations in UK Biobank v2.

Forty-one of the 71 potential novel associations reaching $FDR < 0.1$, including 8 of the 9 novel associations with study-wide significance, were with phenotypes also tested by Neale et al. through GWAS in the expanded UK Biobank ("v2") cohort with up to 337,199 participants of European ancestry. In order to independently replicate our findings, we performed weighted z-score-based meta-analyses between the 23andMe, FINRISK and CHOP PheWAS results, the published GWAS results and the UK Biobank v2 results (excluding the Gplc UK Biobank results). Out of the 41 putative novel associations, 16 showed $P < 0.05$ in UK Biobank v2 with consistent direction of effect, thus independently validating and further strengthening significance of our previous results (Supplementary Table 6). An additional seven potential novel associations showed increased significance in meta-analysis despite $P > 0.05$ in UK Biobank v2, largely due to small number of cases and lack of statistical power in UK Biobank v2 alone. Overall, replication analyses strengthened all eight novel associations with study-wide significance after Bonferroni correction and 23/41 (56%) of the potential novel associations observed in our study, including 8 associations that had been based on results from only a single PheWAS cohort. Strengthened associations in the meta-analysis with UK Biobank v2 include the rs17724992-high blood pressure association that showed significant heterogeneity between the 23andMe and the interim UK Biobank cohorts ($P_{23andMe}=6.4 \times 10^{-10}$, $OR_{23andMe}=0.96$; $P_{UK\ Biobank\ v2}=4.4 \times 10^{-5}$; $P_{meta\ v2}=3.9 \times 10^{-13}$)."

Methods section , I.538-555:

"Meta-analysis with association results from the expanded UK Biobank cohort

To further test the robustness of the putative novel associations identified in our study, we performed a meta-analysis of the 23andMe, FINRISK, CHOP and published GWAS results for 41 SNP-phenotype pairs with association results released by Neale et al. from an analysis of the expanded UK Biobank cohort, consisting of up to 337,199 participants of European ancestry (referred to as "UK biobank v2"). In order to meta-analyze these UK Biobank v2 results, which had been obtained using linear regression models, with the PheWAS cohorts and GWAS results of the current study, which were obtained using logistic regression models, we performed a weighted Z-score meta-analysis. For each SNP-phenotype pair in each study i , we defined weights using the following equation:

$$W_i = 1/\sqrt{(1/N_{a_i} + 1/N_{u_i})}$$

where N_{a_i} and N_{u_i} are the numbers of cases and controls in study i , respectively.

For each SNP-phenotype pair, we then calculated the meta-analysis Z-score as follows:

$$Z = \frac{\sum(W_i * Z_i)}{\sqrt{\sum(W_i^2)}}$$

where Z_i is the Z-score in study i , derived from the logistic or linear regression model.

The UK Biobank GWAS results used in this analysis have been released by the Neale lab under the following URL:

<https://sites.google.com/broadinstitute.org/ukbbgwasresults/home?authuser=0>."

3. Rebuttal accepted, though it highlights the number of independent hits (and tests) to be smaller than claimed, thus the FDR threshold is invalid and should be recomputed.

The reviewer accurately points out that our effective number of tests is smaller than the total number of tests because of the correlation structure of the tests. Dependency of tests and its effect on multiple testing comparison correction has been extensively studied, and Benjamini & Yekutieli (2001) found that FDR is robust to positive dependency amongst tests (specifically the PheWAS application similar to “Problem 4” in the paper) and therefore appropriate for utilization in this study. This is exactly the reason why most current PheWAS use FDR to determine significance, as this procedure 1) better accounts for the structure in the data (i.e. the presence of tests that are not independent) and 2) better performs in the case of hypothesis-driven testing, which is the case in PheWAS where SNPs known to be associated with one or more phenotypes are tested for pleiotropic effects (Glickman et al. 2014. PMID: 24831050). In our study, we further increase the stringency of statistical significance and in **Table 1** only report findings that reach Bonferroni correction ($P < 1.8e-6$), although this cutoff is likely too conservative for the reasons listed above. We agree and discuss (**I.360-363**) that methods will need to be developed to better account for phenotype correlations and to determine the most appropriate significance thresholds for PheWAS.

To clarify this point, we have added the following statement to the Methods section, **I.523-536**:

"It is clear given the structure of this PheWAS and meta-PheWAS that the 27,762 tests are not independent tests, which requires thought about the most appropriate method to control for multiple testing correction. We have chosen two methods, one that provides an extremely, over-conservative multi-testing correction assuming independence (Bonferroni correction) and one less conservative method that has been shown to be robust to test dependency (Benjamini & Hochberg's False Discovery Rate (FDR))⁶⁴. Benjamini & Yekutieli (2001) illustrated that the FDR procedure is robust to positive correlation amongst tests⁶⁵, therefore we have chosen to use the standard Benjamini & Hochberg FDR procedure implemented in the p.adjust method in R. For defining significance in this study, we set a FDR threshold of 0.1, which corresponded to a p-value of $p < 7 \times 10^{-4}$. The over-conservative significance threshold based on Bonferroni correction was $P = 0.05/27,762 = 1.8 \times 10^{-6}$. We note that Bonferroni correction ignores the correlation structure between the tested phenotypes or the fact that all the SNPs tested in this study are known to be associated with one or several phenotypes in published GWAS."

4. Rebuttal accepted

5. Minor comments – rebuttal accepted

Reviewer #3 (Remarks to the Author):

I have reviewed the response and changes made to the manuscript and supplementary materials. The authors have addressed each of my concerns and improved the manuscript overall. The addition of Supplementary Figures 2 and 3 are particularly insightful.

Response to Reviewer 3:

We are glad that we were able to address the reviewer's comments.

Additional revisions:

*Since we submitted the manuscript and during the review process at Nature Communications, a new T2D exome chip study was published in April 2018 (PMID:29632382), reporting a significant association between *PNPLA3* rs738409 and T2D. Consequently, we have removed the rs738409-T2D association from the list of novel associations in our study, bringing the numbers of potential novel associations to 9 associations reaching study-wide significance after Bonferroni correction, and 71 novel associations reaching FDR<0.1.

Changes made to the manuscript are indicated by underlined text.

Phenome-wide association studies (PheWAS) across large population cohorts support drug target validation

Reviewer 2, round 3:

I am disappointed to see that the authors are not addressing the persistent issues with the paper, comments 1a and 2.

Comment 3 is addressed in full – I indeed mis-stated FDR, and I stand corrected.

Comment 1b is addressed through rebuttal iii, though a p-value for the purported R^2 value should be listed in the paper, along with a supporting p-value. Rebuttal i basically agrees with my statement that phenotypes from only CHOP are not relevant to the issue of including it for other phenotypes. Rebuttal ii is not rebutting anything. The opening sentence agrees with me. Since rebuttal iii works, it is not a problem that rebuttals i and ii don't.

Comment 1a still stands:

Rebuttal i is not valid.

The QC steps of the individual studies do not address population issues when combining them, e.g. when artifacts are small enough to require correction at each individual study but become problematic in terms of their magnitude upon combining the studies.

Rebuttal ii is not valid.

The authors were offered a way out, to revise their power estimation, that I assume is over optimistic. Yet, the authors dug in their heels, and claim power to replicate is high, as advertised. If this is the case, and power is 1.0 for a SNP to replicate, then it must replicate. The authors cannot find solace by a majority of their finding replicating. Any SNP with power of 1.0 means it has to replicate. If it doesn't, something is wrong. I am not reviewing Denny et al., which might be underpowered to replicate many SNP, therefore could in principle be fully self-consistent with lower replication rate. This study is not.

Comment 2 still stands.

I applaud the authors for attempting replication in UK biobank V2. They indeed succeed replicating 16 of the 41 results achieving $FDR < 0.1$. Since 4.1 of these are expected to be false, that's 11.9 expected replicated results. This is good news. Also one of 8 novel associations reaching Bonferroni correction is replicated. That's also good news. The rest are not, and I require this to be clearly stated in the abstract, to separate true results from unsupported ones.

Reviewer #2 (Remarks to the Author):

The authors were offered a way out, to revise their power estimation, that I assume is over optimistic. Yet, the authors dug in their heels, and claim power to replicate is high, as advertised. If this is the case, and power is 1.0 for a SNP to replicate, then it must replicate. The authors cannot find solace by a majority of their finding replicating. Any SNP with power of 1.0 means it has to replicate. If it doesn't, something is wrong. I am not reviewing Denny et al., which might be underpowered to replicate many SNP, therefore could in principle be fully self-consistent with lower replication rate. This study is not.

Response to Reviewer #2's comment:

The assumption that any SNP with a power of 1.0 must replicate might be correct if discovery and replication cohort fully corresponded for all relevant parameters. GWAS and real-world data cohorts clearly do not. They are ascertained under very different settings, and that RWD cohorts have at all the potential to be used for meta-analyses with GWAS data across a broad range of phenotypes and still be relevant for drug development is among the key messages of our paper. Any power estimates for replication are approximations and among others heavily depend on the accuracy of effect size estimates in the discovery cohort, as well as non-adjustable, cryptic differences between discovery and replication cohort composition and endpoints. That replication does not always reach the predicted power can be used to hypothesize what might contribute to such cryptic differences. We do so e.g. in lines 161-166 and 1.383-385 for inflammatory bowel disease (IBD) and its subtypes Crohn's disease (CD) and ulcerative colitis (UC), which constitute 8 of the 11 associations that failed to replicate, although directionality in our meta-PheWAS matches that of previous GWAS.

We note that the replication rates of 66% in PheWAS from Denny et al. (see our previous response to the reviewer) had been calculated using the same criteria as in our study.